# Socioeconomic predictors of vulnerability to flood-induced displacement

**Benedikt Mester** [1,5], **Katja Frieler** [1,2], **Oliver Korup** [2,3], **Bina Desai** [4] & **Jacob Schewe** [1] ✉

Floods displace an average of 12 million people every year, and are responsible for 54% of all disaster-induced displacements. Displacement risk scales with the vulnerability of exposed populations, but this vulnerability is poorly understood at a global scale. Here we show that measures of human development and rural areas explain more of the variance of displacement vulnerability than income levels measured by gross domestic product. We combine global flood and displacement data to estimate vulnerability, as the ratio of displacement to exposure, for over 300 historical flood events. We find that this vulnerability varies by several orders of magnitude both between and within countries. A random forest regression shows that infant mortality rate and population density are among the most important predictors of displacement vulnerability at national level and within countries, respectively, highlighting the vulnerability of low-income and marginalized populations and of rural communities. Our results indicate that, rather than relying on overall economic development alone, targeted investments are needed to improve living conditions and coping capacities for the most vulnerable groups, particularly outside of large cities, and to prepare for increasing flood hazards due to climate change.

More than 195 million people worldwide have been displaced by floods since 2008. This is more people than by any other type of disaster, and more than by conflicts and violence[1] (https://www.internal-displacement.org/database/). Disaster displacement refers to situations in which people are "forced or obliged to flee or to leave their homes or places of habitual residence, in particular as a result of or in order to avoid the effects of [...] natural or human-made disasters"[2]. Displacement avoids fatalities, but disrupts livelihoods, undermines well-being, and incurs substantial costs on communities and countries[3]. Displacement risk is a product of the physical properties of flooding (hazard), the exposure of people, their assets and livelihoods to flooding, and vulnerability, i.e., the susceptibility and lack of resilience to being displaced[4,5]. Flood hazard has been changing over the past five decades[6–9] and, in

particular with respect to rare, large events, is expected to increase in many regions under continued climate change[10–12]. More than 20% of the world's population are currently exposed to high flood risk[13], and population growth and urbanization are set to raise this exposure further, particularly in lower-income countries[14,15]. At the same time, progress in reducing vulnerability has not been sufficient to reduce overall disaster risk[16,17]. Against this backdrop, it is important to understand and quantify flood displacement vulnerability. Knowing what determines this vulnerability is important for understanding past trends in displacement risk; anticipating future changes; and identifying entry points to improving the resilience of affected communities to reduce displacement risk.

However, it is unclear how displacement vulnerability varies between flood events, and which factors, beyond differences in

[1]Potsdam Institute for Climate Impact Research (PIK), Member of the Leibniz Association, Potsdam, Germany. [2]Institute of Environmental Science and Geography, University of Potsdam, Potsdam, Germany. [3]Institute of Geosciences, University of Potsdam, Potsdam, Germany. [4]CGIAR Climate Security, Alliance Biodiversity International and CIAT, Rome, Italy. [5]Present address: Swiss Re, Arabellastraße 30, Munich, Germany. ✉e-mail: jacob.schewe@pik-potsdam.de

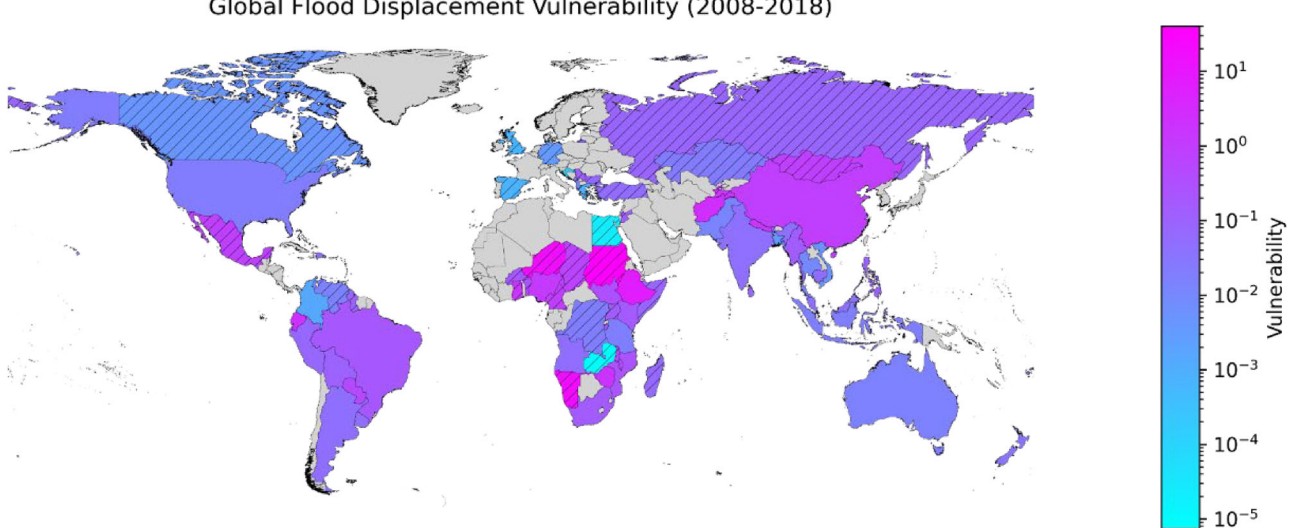

**Fig. 1 | Country-level median vulnerability to flood-induced displacement.** Vulnerability is obtained for each flood-displacement event as the ratio between estimated displacements and estimated population exposure; the median is then calculated from all available events in a country. Gray color indicates countries with no events. Hatching identifies countries with less than three reported events. Country borders are derived from GADM[65].

hazard and exposure, might explain this variation. Only few studies have explored flood-induced displacement at the global scale[18–21]. Displacement is mostly low in countries with gross national income above $13k (2020 international $) per capita, while both low and high rates of displacement (per country population) are observed across lower-income countries[21]. It remains unresolved how much of the variation in displacement can be attributed to national income levels. Similarly, little is known about the role of non-economic or local factors, such as urban development and infrastructure access, demographics, or social disparities, which are important drivers of social vulnerability to flooding in many case studies[22–24] and large-scale assessments[25] but have rarely been considered in the context of displacement.

Here we combine reported displacement data with remote-sensing data of flood extents and gridded population estimates, to estimate vulnerability, as the ratio between displacement and flood exposure, for over 300 large fluvial and coastal flood events that occurred around the world between 2008 and 2018. We examine which predictors, measured at sub-national resolution, explain most of the observed variation in displacement vulnerability between individual events, using a mixed-effects random forest[26] to account for unobserved country-specific factors (i.e., average vulnerability might be lower in one country than in another). To gain insight into these potential country-specific factors, we apply random forest regression to predict the median vulnerability per country using predictors measured at the national level. While vulnerability is ultimately relevant at the local level, it is impossible to directly measure all its possible determinants across many countries. Factors such as the presence of disaster early warning systems, physical protection measures, the availability of emergency and recovery assistance, or public awareness to flood hazards are hardly documented at global resolution. Such elements may, however, be reflected by national-level characteristics such as public assets or forms of governance. Hence, country-level indicators might explain some of the variation in vulnerability across countries as opposed to indicators only available at the local level.

Our combination of the best available global flood observation data with the most complete and detailed global displacement estimates is unique compared to previous global studies of flood vulnerability[21,27,28]. Our main methodological choices are motivated as follows. First, we use remote-sensing, rather than modeled, flood hazard data to warrant consistency and avoid model uncertainty[29],

providing more accurate exposure estimates for each flood. Second, we use geocoded displacement information on level-1 or level-2 subnational administrative units (e.g., provinces or districts) for a finer resolved analysis than at national level[21]. Thus, we can identify the local context of displacement events, and address variations in displacement vulnerability not only across, but also within, countries. Third, as opposed to many previous studies that have focused on single predictors of flood impacts, such as national income or population size, we choose a multivariate analysis. Drawing from a larger set of plausible predictors, and using random forest regression, our analysis can also account for non-linear effects of, and interactions between, these predictors. Finally, using the vulnerability ratio as the target variable controls for the expected close association between exposure and displacements prior to the regressions. This narrows the distribution of the target variable (Supplementary Fig. S1) and makes sure we estimate predictor effects on vulnerability rather than on exposure. The third and fourth aspects especially differ from a recent study that estimated displacements from a smaller number of local-level independent variables in a linear regression framework[18].

## Results

### Global displacement vulnerability
Vulnerability to flood displacement varies between countries by several orders of magnitude (Fig. 1, and Supplementary Fig. S2). It is high (>0.1, meaning one or more displacements for every ten people exposed) in many South American, Sub-Sahara African, and Asian countries. Countries with high vulnerability include Ecuador, Ethiopia, Zimbabwe, Nigeria, Afghanistan, Nepal, and China (Supplementary Fig. S2); some countries have only one or two data points (Supplementary Fig. S3). We estimate median vulnerability of >1 across multiple events) in several countries including Afghanistan, Ecuador, and Ethiopia, while vulnerability to individual events was >1 in two thirds of countries with at least three reported flood events (Supplementary Fig. S2). Formally, vulnerability expresses a fraction of loss and thus cannot exceed 1. However, our estimate of vulnerability may exceed 1 if our estimate of displacements exceeds our estimate of exposed population. This can be due to preemptive evacuations[30], which cannot be separated from post-disaster displacement in the data; or due to social dynamics that displace people even when they are not personally exposed. This may happen if people follow their kin, or if their places of work or other source of

**Table 1 | Explanatory power ($R^2$) of the five preferred predictor combinations**

| Predictor 1 | Predictor 2 | Predictor 3 | $R^2$ |
|---|---|---|---|
| log(GDPpc) | log(Elevation) | log(Population Density) | 0.31 |
| Event duration | log(Elevation) | log(Population Density) | 0.30 |
| log(Elevation) | Population ≤14 years | log(Population Density) | 0.28 |
| log(Elevation) | log(Population Density) | NaN | 0.28 |
| log(Elevation) | FLOPROS | log(Population Density) | 0.27 |
| **Predictor 1** | **Predictor 2** | **Predictor 3** | **$R^2$** |
| Mortality rate, infant (per 1,000 live births) | Educational index | Urban population (% of total population) | 0.34 |
| Population density (people per sq. km of land... | Educational index | Urban population (% of total population) | 0.31 |
| GDP per capita, PPP (current international $) | Educational index | Urban population (% of total population) | 0.30 |
| Educational index | Urban population (% of total population) | NaN | 0.29 |
| Population ages 0-14 (% of total population) | Educational index | Urban population (% of total population) | 0.28 |

Event-specific vulnerability using mixed-effects random forest (top); country-level median vulnerability using random forest (bottom). "-" indicates that the model has no third predictor

income or important infrastructure, such as schools or childcare, suffer damage[31,32]. Vulnerability estimates > 1 may also reflect an over-reported displacement, or an underestimated exposure. An underestimated exposure could in turn arise either from incomplete space-borne flood extent observations[33] (for instance, small but important features such as flooded streets in urban areas may not be captured) or low-quality population data[34]. There is no significant trend in global median vulnerability over the period 2008-2018 (Supplementary Fig. S4). Given that event-specific vulnerabilities vary by orders of magnitude even within countries, we use $\log_{10}$-transformed values[35]; thus, our analysis concerns the magnitude of vulnerability. This approach also acknowledges the low, if not unknown, accuracy of displacement statistics[36].

**Predictive Models**

To understand the possible determinants of flood-displacement vulnerability, we first select potential predictors based on a review of the literature on flood-related social vulnerability, in addition to physical characteristics of the floods and inundated areas (Methods). A set of up to three such candidate predictors feeds into a random forest regression, excluding combinations of closely related and mutually correlated predictors. We test many different models (predictor combinations) in a leave-one-out cross-validation setup. Our five preferred models (highest $R^2$) in the event-level analysis have $R^2$-values of 0.27-0.31, and all include population density and elevation as predictors (Table 1). Ranking models by the Akaike information criterion (AIC) or Bayesian information criterion (BIC), which penalize models with more predictors, yields nearly identical results as ranking them by $R^2$ (Supplementary Table S1). In the country-level analysis, our five preferred random forest models have $R^2$ values of 0.28 – 0.34, and all include the level of urbanization (share of urban population in total population) and education index as a predictor. Again, ranking models by AIC or BIC yields very similar results (Supplementary Table S2). The modest $R^2$ values, while not surprising given the complexity of the issue, mean our models only partially explain vulnerability. The predictor importance ranking discussed below must be viewed in this context of low explained variance; nevertheless, they provide meaningful insights into the relative roles of different socioeconomic factors.

**Important predictors at the event-level**

To assess the importance of an individual predictor across different models, we test all models containing the predictor of interest before and after randomly permuting its measurements, and compare the resulting $R^2$ values. The change in $R^2$ due to randomization is a measure of the predictor's contribution to the model skill, also termed feature importance[37]. We rank predictors by the median decrease in $R^2$ after

randomization (Fig. 2). Alternatively, predictors can also be ranked by the median $R^2$ across all models containing the relevant predictor (Supplementary Fig. S5). While this measure does not treat individual predictors entirely independently, it results in a similar ranking of the most important predictors as that by decrease in $R^2$. Results are also very similar when we rank predictors by the median increase in AIC (Supplementary Fig. S6) or BIC (Supplementary Fig. S7) after randomization, compared to ranking by decrease in $R^2$.

The event-specific analysis indicates population density and elevation as the most important predictors (Fig. 2, top). This result is consistent with the widespread presence of population density and elevation in the models with the highest $R^2$ (Table 1). The ranking also shows that these two predictors are more important than GDP per capita. This finding is crucial, because GDP per capita or some related measure of income levels is often used as the single indicator of socio-economic vulnerability, and assumed to be a reasonable proxy of measures of social status, economic deprivation, etc[21,38]. Our results show that the variance in flood-displacement vulnerability is better explained by factors other than aggregate income levels (as measured by GDP per capita) alone (Fig. 2, top). These results are robust when using the ratio of deaths to exposure as an alternative target variable (Supplementary Fig. S8), suggesting they may represent general aspects of flood vulnerability.

We show the marginal effect of a given predictor on flood-induced displacement vulnerability in partial dependence plots. We find that population density has a negative marginal effect (Fig. 3). All else being equal, places with low population density are associated with high flood-displacement vulnerability. Our findings thus indicate that sparsely populated, rural areas tend to be highly vulnerable on average. This observation is consistent with theories and individual empirical studies on rural and urban flood vulnerability[22,24,39,40] that point out high vulnerability in rural areas. Our results support this notion systematically in relation to displacement, for a global context with a large number of observations. We recall that the extent of urban floods may be underestimated e.g., when short-lived or small features such as flash floods or flooded streets are missed by satellite imagery[33]; however, such a bias would imply that we overestimate vulnerability in urban areas, and thus our finding of higher vulnerability to displacement from fluvial and coastal flooding in rural areas compared to urban areas remains robust.

Vulnerability to floods and other disasters can be larger in rural areas than in cities for physical but also social and economic reasons[22]. In physical terms, small rural communities may have a much larger share of their population or assets exposed to a given hazard than large cities. For fluvial and coastal floods, in an urban context, much of the population living in the area for which exposure and vulnerability are assessed (e.g., some administrative unit or a grid cell) may be less

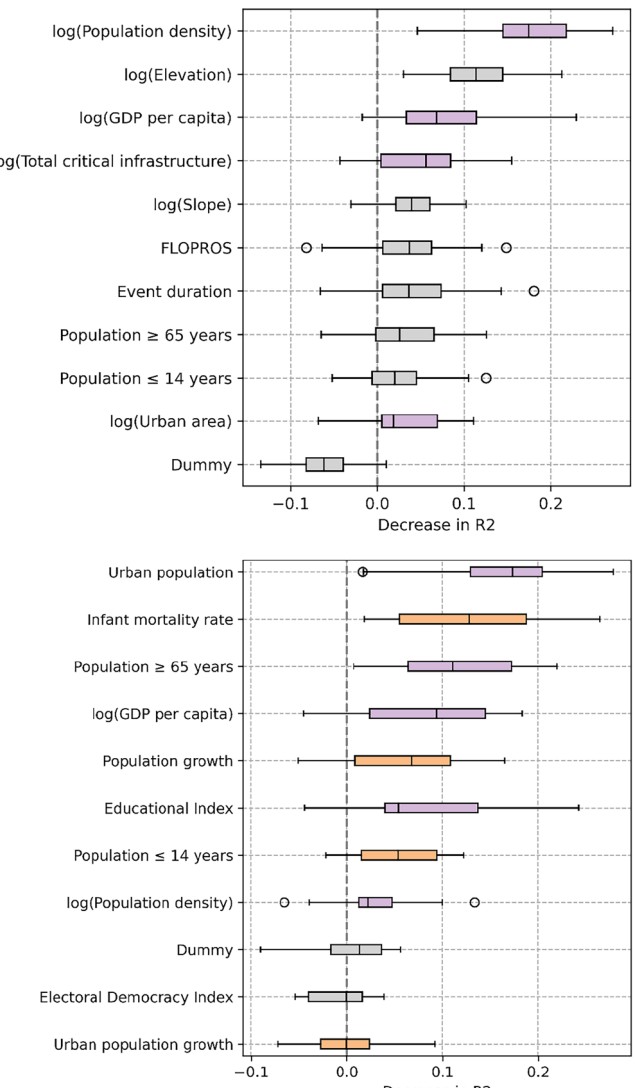

**Fig. 2 | Feature importance in the event-level analysis (top; $n$ = 303) and the country-level analysis (bottom; $n$ = 72), measured by the median decrease in $R^2$ after randomizing.** Results relate to the test data using leave-one-out cross-validation. Each box plot represents all models using the predictor of interest; up to three predictors are used per (mixed effects) random forest model. Vertical line, box, whiskers, and circles indicate the median, interquartile range, 1.5 times the interquartile range, and outliers, respectively. Only models with $R^2 \geq 0.01$ are included. Box color indicates whether an increase in the predictor value is estimated to have an increasing (orange), decreasing (purple), or non-monotonic or ambiguous effect (gray) on vulnerability (cf. Figures 3 and 4).

exposed to hazardous or damaging water levels e.g., because of variations in elevation across the city, and multi-story residential buildings or other infrastructure may provide refuge and prevent displacement. In contrast, a small village may get completely flooded quickly, offering little for its inhabitants to take refuge, and making it much more likely that most or all of its population may be displaced. These physical aspects concern fine-scale variations in exposure, which our data cannot distinguish, and which are thus subsumed in our vulnerability metric. In terms of social and economic reasons, rural areas tend to be relatively poorer, with lower structural resilience of buildings, and to be neglected or treated subordinately by centralized government, resulting in higher vulnerability against floods and other disasters. For example, levees that protect larger settlements may lead to even higher flood levels for neighboring or downstream, smaller settlements. Rural areas may also lack economies of scale, as cities can

afford much larger emergency response capacities such as professional fire brigades[22,39]. Resilience against floods and other disasters differs markedly between rural and urban counties in the USA[39]; such differences are likely to be more pronounced in less wealthy countries.

The marginal effect of the second most important predictor, elevation, is largely positive, such that vulnerability increases with elevation. Floods in mountainous regions tend to have different properties than floods in low-lying areas, for example mainly higher velocity of flow, or potentially damaging debris carried by the water[41,42]. At the same time, mountain regions often have a different socioeconomic structure than lowlands: infrastructure and economic development are heavily modulated by topography; and a common demographic pattern is that young people move to cities while older people remain in mountain villages and towns[43,44]. The increased vulnerability at higher elevations may thus partly reflect differences in age, educational, and economic characteristics influencing vulnerability and adaptive capacity. The partial dependence plot also indicates increased vulnerability at very low elevations, although this observation is based only on few samples and may be less reliable (Fig. 3). These areas below -10 m above sea level are mainly coastal areas which globally are often densely populated and susceptible to coastal flooding; with their flat terrain, they may be associated with longer flood duration on average than more rugged areas.

The effect of (sub-national) GDP per capita, which is ranked third most important predictor by decrease in $R^2$, is negative in the range of about $ 6k to 10k (2017 PPP), while there seems to be little effect at either lower or higher income levels. While the cited range corresponds to the highest data density, many data points are available between about $ 1.3k and 20k, supporting a non-linear marginal effect. This means that high-income places tend to be less vulnerable than low-income places, but there is a lot of variation in vulnerability in both the low-income and the high-income range unexplained by income levels as measured by GDP per capita. Critical infrastructure has a negative marginal effect on displacement vulnerability, consistent with studies showing high flood vulnerabilities in undersupplied, informal settlements[24]. The remaining predictors show mostly small or indeterminate marginal effects (Fig. 3 and Supplementary Fig. S9), which is consistent with their low feature importance ranking. This includes a measure of flood protection standards (FLOPROS), which in our context does not represent the effectiveness of flood prevention (we only study floods which were not prevented) but the possibility that higher flood protection standards may also be associated with stronger flood emergency response capacities. However, according to our analysis, this measure is of low importance in explaining displacement vulnerability; which may also be related to the high uncertainty of protection standard estimates in many parts of the world[45].

## Important predictors at the country-level
At the country-level, urbanization level and infant mortality rate are the most important predictors, ranked by decrease in $R^2$; followed by the share of elderly population (65 years and older) and GDP per capita (Fig. 2, bottom). When instead using the increase in AIC or BIC, the relative ranking of these predictors changes slightly, with education level and share of elderly population ranked more important than infant mortality. In any case, urbanization level, infant mortality rate, and share of elderly population are ranked more important than GDP per capita. While these non-economic, human development-related factors are linearly correlated with GDP per capita ($r$ = 0.81 for urban population, and $r$ = 0.91 for education; Supplementary Fig. S10), the finding that they are more weighty predictors suggests they contain important information related to the causes of vulnerability. For instance, most causes of infant death are preventable with low-cost measures[46], thus infant mortality rate is a measure of human development that is sensitive to deficient healthcare (and by extension, generally inadequate living conditions) even in a small fraction of a

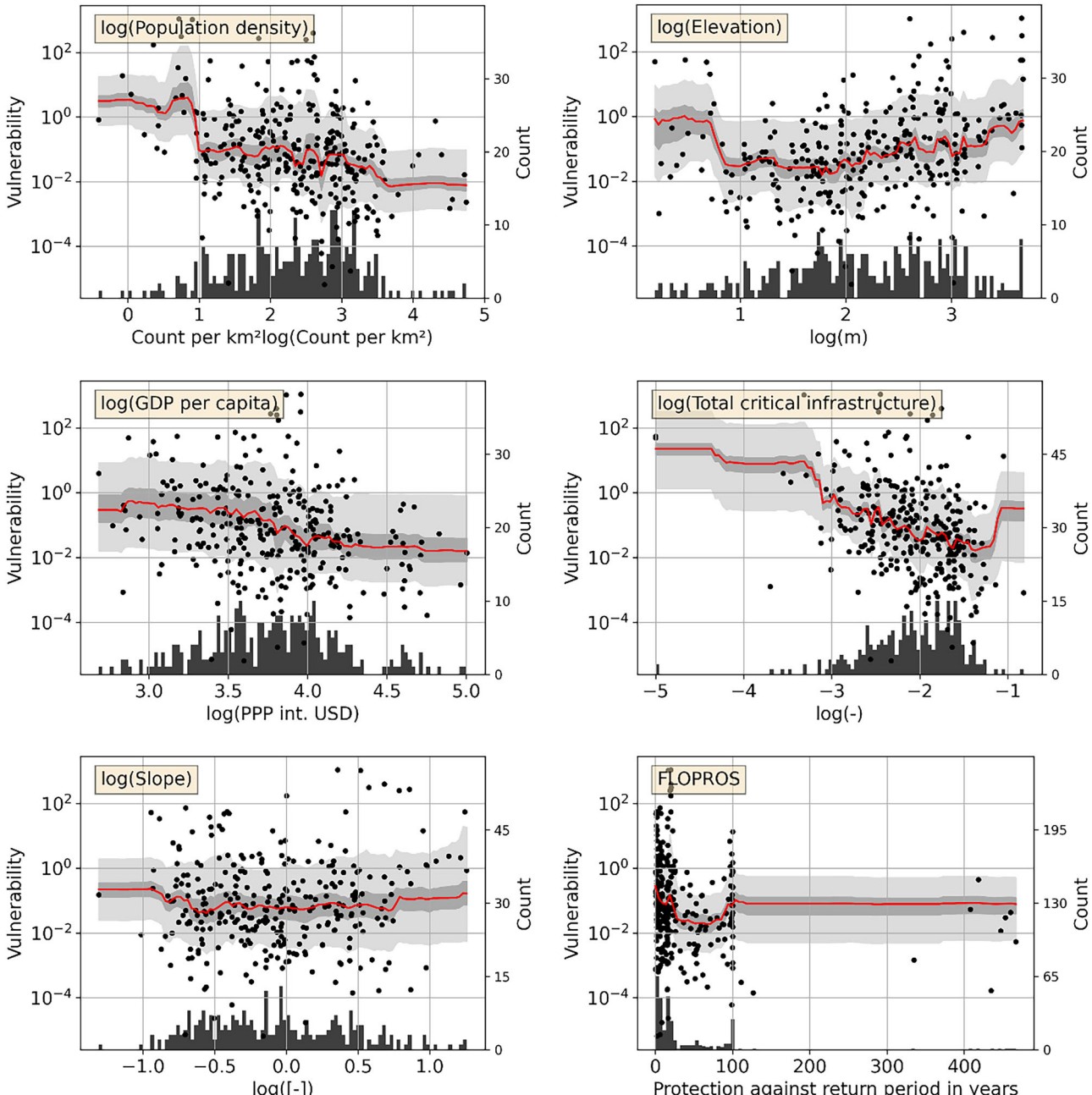

**Fig. 3 | Partial dependence plots (PDPs) for the most important event-level predictors.** Each PDP uses the model with the highest drop in R² after randomizing the predictor of interest. The median vulnerability per predictor increment is indicated in red, uncertainty intervals are in light gray (5th/95th percentile) and dark gray (33th/66th percentile). Black dots show observations (vulnerability per documented flood), and histograms (secondary *y*-axis) show the data distribution. This figure shows the four most important predictors each by decrease in R² and by total R² (population density and elevation are among the four most important predictors according to both rankings, therefore a total of six predictors are shown here). PDPs for the remaining predictors are in Supplementary Fig. S9.

country's population; whereas in country-level GDP per capita, income differences within the country get averaged out. The result that GDP per capita is not the most important predictor at sub-national level either suggests that, for similar reasons, aggregate income levels–at least as measured with available data products–may inappropriately capture vulnerability even when averaged over smaller areas.

The age structure variables (population aged 14 years and below, and 65 years and above, respectively) were also included in the event-level analysis, but were of relatively low importance there, while they are more important at the country-level. The same holds for urbanization, expressed by urban area at the event-level, and by the share of urban population at the country-level. These variables may thus be more indicative of the overall level of development and vulnerability in a given country or region, while other factors play a more important role in explaining the local variation between different events within a country. In particular, while the share of urban area at subnational level is only moderately correlated with population density (Supplementary Fig. S10), national-level urbanization is indicative of the overall fraction of population living in rural settings (corresponding to low population density at a subnational level), and thus being potentially more vulnerable, along the lines described in the previous section.

Partial dependence plots for the most important predictors in the country-level models show that urbanization and the share of population aged 65 years and older both have a negative marginal effect on

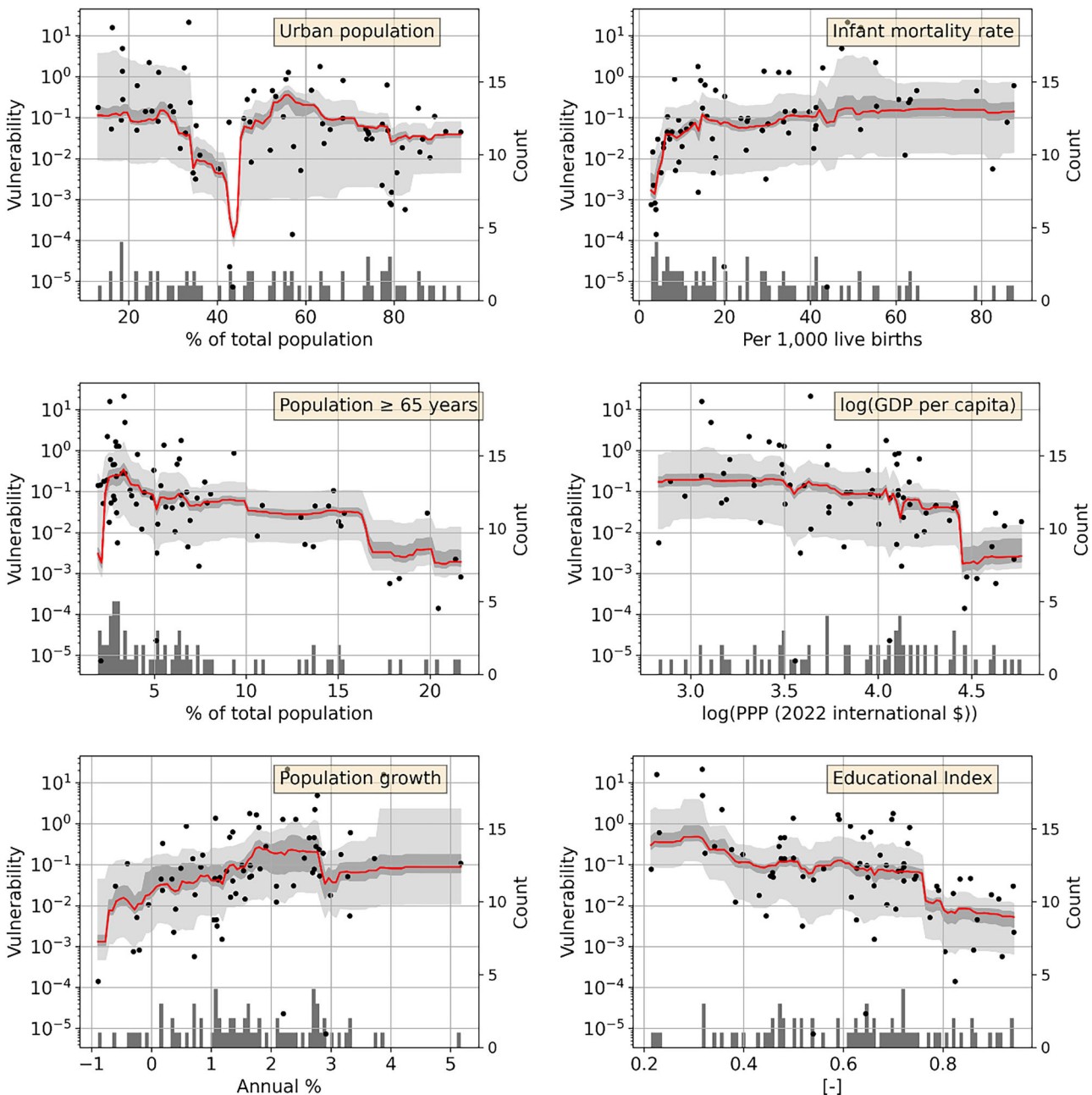

**Fig. 4 | Partial dependence plots for the most important country-level predictors.** Symbols and uncertainty bands as in Fig. 3. This figure shows the six most important predictors by decrease in R², which include the four most important predictors by total R². PDPs for the rest of the predictors are shown in Supplementary Fig. S9.

vulnerability. In other words, vulnerability tends to be high in less urbanized countries and countries with a small proportion of elderly population. The first aspect may be related to the observation that in countries with low urbanization levels, relatively many people (compared to highly urbanized countries) live in rural areas which tend to be more vulnerable[40] - linking back to the importance of population density in the subnational models. A high share of elderly population is related to a high life expectancy, which in turn is an indicator of human development and, more specifically, well developed health care systems and other social services[47–49]. In contrast, a high share of young population is often related to poverty and low levels of access to social services[24,50,51]. Regarding infant mortality, predicted vulnerability is low only for very low infant mortality rates, but consistently high for infant mortality rates from around 1%−8% (Fig. 4). This suggests the capacity to prevent infant deaths is a strong indicator of broader societal

development. The nonlinear shape of the PDP also confirms the importance of using flexible methods, such as random forest, that do not impose a linear relationship.

The level of education (as measured by mean current and expected future years of schooling) has a clear negative marginal effect on vulnerability, while the population growth rate has a positive effect (Fig. 4). GDP per capita, which on the country-level is the fourth most important out of ten predictors, shows only a slight negative marginal effect on vulnerability at low and intermediate GDP values, while at higher values, vulnerability is predicted to be more significantly lower (Supplementary Fig. S9). This is in agreement with the observation that vulnerability is generally low in most high-income countries, whereas both low and high vulnerabilities are observed in lower-income countries[21]. The relative weakness of this predictor compared to urbanization and infant mortality rate shows that such additional, non-

economic factors might be important in explaining the variance in vulnerability across most low- and middle-income countries.

## Discussion

Vulnerability to flood-induced displacement is poorly understood in comparison to mortality or economic damages induced by flooding. In light of 10 million flood-induced displacements worldwide in 2023 alone[1], a better understanding of vulnerability is important to leverage risk reduction strategies and adaptation planning[4,52]. We estimated event-specific vulnerability values for 303 recent flood events in 72 countries, and found that they vary by orders of magnitude both within and across countries. Particularly high vulnerabilities were estimated in some African countries, such as Ethiopia, Nigeria, and Zimbabwe; but also in China, Nepal, Afghanistan, and Ecuador. High vulnerability is thus widespread among, but not limited to, the lowest-income countries.

We investigated the importance of a range of social, economic, political, and physical factors in explaining variations in the magnitude of vulnerability across events and across countries. Within countries, using mixed-effects random forest models ($R^2 \leq 0.31$), population density emerges as the most important factor, followed by elevation. At the same time, GDP per capita, a commonly used predictor often associated with disaster risk and vulnerability, is relatively unimportant in our models. We have used a recent global reconstruction of subnational GDP per capita[53] which utilizes data on urbanization levels, travel time to the closest city, and national-level income inequality to estimate subnational GDP per capita in countries with no reported data, which concerns many African and some Asian countries.

While sparsely populated, rural areas have been shown to be especially vulnerable to flood-related impacts in empirical and qualitative studies based on data from the USA[39] and across the world[22,24,40], this finding is new in the context of displacement. This result underlines the importance of clearly separating the effect of population density on vulnerability from its role in determining exposure: in densely populated areas, more people will be exposed to a given flood event. Studies trying to predict displacement using population density as an explanatory variable may conflate the two effects and produce biased results. In our study, we predict (the log of) displacement vulnerability as the ratio between displacement and exposure, thus accounting for the effects of population density on exposure. The remaining effect of population density on vulnerability, highlighted by our models, indicates residents of rural areas on average suffer higher displacement risk than their urban counterparts, for a given hazard. Our models estimate a two orders of magnitude difference in vulnerability between sparsely and densely populated areas, all else equal, which would imply a large potential for risk reduction from better protecting rural populations.

Across countries, the most important factors explaining differences in the average order of magnitude of vulnerability, according to random forest models ($R^2 \leq 0.34$), are urbanization and infant mortality rate, followed by the share of elderly population and GDP per capita. When model quality is measured by AIC or BIC instead of $R^2$, education is similarly ranked very highly. These observations are in line with general theories stating that a higher level of human development leads to a decrease in vulnerability[5,54]. Age structure, health, infrastructure, and education indicators are often found indicative of vulnerability to climate extremes for communities and individuals[55]. Infant mortality rate may serve as a proxy for public service delivery in the context of climate extremes, such as emergency and recovery assistance, and has been found to explain parts of the variation in disaster-related deaths also within countries[56]. High infant mortality rates are also related to the marginalization of affected communities[57], which is associated with higher flood risk[58], caused by, for example, a lack of early warning systems, physical protection measures (which are not well measured by FLOPROS in many Global South countries), or

official emergency and recovery support. More directly, infant mortality can contribute to displacement during and after floods by acting as both a health crisis and a socio-emotional tipping point for affected households, compelling them to flee high-risk zones. In regions where floodwaters damage healthcare infrastructure or impede access to essential services, the elevated risk of infant death due to preventable causes—such as waterborne diseases or lack of neonatal care—can push families to migrate in search of safer environments with better medical access.

Again, we find that GDP per capita is relatively unimportant in the country-level models. GDP per capita, though widely used to characterize socioeconomic status, may thus be a poor measure of vulnerability to flood-induced displacement. One reason may be that communities are very heterogeneous, and the lowest-income and most vulnerable parts of the population may not show up in average income levels as measured by GDP per capita, whereas factors like infant mortality are sensitive to the presence of marginalized, impoverished, or undersupplied subpopulations. The marginal effect of the infant mortality indicator on vulnerability is highly nonlinear, highlighting the usefulness of models like random forest that do not impose a-priori linear relationships. For the real world, our results suggest reducing poverty and improving living conditions for the most vulnerable parts of the population might, among countless other benefits, effectively reduce displacement risk.

While random forests are well-suited for small samples[59], the limited amount of data available for our analysis introduces considerable variance in our results, which prevents us from drawing more detailed conclusions. Similar to other studies, a lot of residual variation in vulnerability remains that we cannot explain with our models, with $R^2$ values rarely exceeding 0.3. This variance may be partly related to data quality issues regarding the measurement of displacement. While IDMC to our knowledge is the highest-quality global source of displacement data, no uncertainty estimates are available for their figures, and IDMC figures and those from another global data provider (the Dartmouth Flood Observatory, DFO) can differ by several orders of magnitude (Supplementary Fig. S11). That said, some of our main findings—the prime importance of population density in the event-level analysis; the role of infant mortality rate, urban population, and share of elderly population among the most important predictors in the country-level analysis; and the relatively lower importance of GDP per capita—are robust even when DFO displacement data are used instead of IDMC data (Supplementary Fig. S12 and S13; note $R^2$ values are lower with DFO data than with IDMC data, Supplementary Fig. S14–S16).

On the other hand, the unexplained variance reflects to some degree the limited understanding of the many processes that shape vulnerability at the community and household level. Consequently, our models provide useful insights into some of the factors explaining variations in displacement vulnerability; they are less skilled for reliable prediction of displacement from any given flood event. Nevertheless, our results go beyond those of recent studies employing random forest[19] or negative binomial[18] regression to predict displacement (rather than displacement vulnerability). Ronco et al.[19] attributed displacement mainly to a combination of intense precipitation and poor household conditions, albeit without a direct measure of flooding. Our study instead controls for the magnitude of flooding induced by precipitation and other drivers, and the importance of the infant mortality rate in our models is consistent with the importance of related measures of poor household conditions. However, we reveal the increased vulnerability of rural communities measured through the population density indicator. Vestby et al.[18] found national income, level of democracy, and conflict as important factors explaining some of the variation in displacement magnitude. In comparison, we find that other factors are more important than national income, and these factors do not include the level of democracy which ranks very low in our study. Moreover, Vestby et al.[18] found extreme displacement

positively associated with nighttime luminosity, which they interpreted at least partly as population exposure. Our study explicitly controls for population exposure to flooding in the definition of the target variable, and therefore our findings can more readily be interpreted as relating to the importance of a given predictor in terms of vulnerability.

A negative relation between the economic status and vulnerability in terms of fatalities and economic losses, was found in previous works for a set of hazards, including flooding[60]. Mortality rates are relatively high among countries with less than US$ 10k per capita, while a vulnerability threshold for economic loss rates varies between US$ 10k and US$ 15k[27,28]. Globally, gross national income per capita is negatively associated with flood-induced displacements per 1000 people, with a breakpoint at US$ 13k per capita[21]. We find a nonlinear negative relationship between GDP per capita and displacement vulnerability at the event-level, with the strongest decline in vulnerability between ~$ 6k and 10k (2017 int. $ PPP), as well as at the country-level, with a drop at approximately US$ 28k (2022 US$ PPP). Thus, on the one hand, our study confirms a negative and non-linear association between aggregate income levels and vulnerability also with respect to displacement. On the other hand, it shows that other, not directly economic, factors are more important in explaining observed variations in the magnitude of displacement vulnerability.

Major challenges in understanding displacement vulnerability are the limited quality and granularity of the displacement data (Buhaug, 2023). Conceptual uncertainties, practical difficulties involved in collecting displacement data on the ground, and potential misreporting by media sources, mean that reported numbers are unlikely to be accurate. While the uncertainty is hard to quantify, the order or magnitude of displacements, which we have addressed in our analysis, may be a more reliable indication than the absolute number of displacements. Moreover, the available data fail to distinguish post-disaster displacement from preemptive evacuations. While the latter can save lives, both forms of displacement are associated with costs and burden[30]. Our results do not appear to be primarily driven by the presence of preemptive evacuations in the data: sparsely populated areas, identified as particularly vulnerable in our study, are less likely to receive early warnings and to be targeted for planned evacuation than urban centers; and countries with high infant mortality rates are less likely to have the financial and institutional resources required for organizing effective evacuation campaigns. Disaggregated estimates by the Internal Displacement Monitoring Center (IDMC)[61] for 2023 show that out of 3180 flood-induced displacement events in that year, only 46 were associated with preemptive evacuations, while 2171 were associated with post-disaster displacement (963 events had no related information). This record indicates that preemptive evacuations may constitute a minor fraction of all displacement events. Nevertheless, the missing distinction between post-disaster displacement and preemptive evacuations is a major caveat of our study, and may put important limits on the overall explanatory power of our models.

Many global studies used intensity-damage functions to represent vulnerability, for example, depth-damage functions are often used for flood risk assessments[62]. Previous studies have shown that flood depth explains a substantial part, but not all, of the variations in flood damages; simple models with only flood depth do not perform well[63]. While flood-depth information is unavailable for the observed flood events studied here, we have included indicators such as elevation, slope, and flood duration in order to characterize some of the physical aspects of the hazard. Our finding that vulnerability tends to be higher in high-elevation and very low-elevation areas may be related to physical characteristics of mountain floods, but also socioeconomic and demographic differences between mountain regions and lowlands. Besides this, our study for the first time sheds light on the role of several social, economic, and political factors, for the variability in flood-induced displacement outcomes between as well as within countries. It thus adds to the literature on the determinants of

vulnerability to climate events[25]. To advance flood-displacement risk assessment, future studies may consider additional aspects of physical vulnerability alongside those of social vulnerability[64].

Our findings suggest that relying on economic growth on its own to reduce vulnerability to flood-induced displacement is delusive. Rather, targeted investments are needed in particular to improve coping capacities in rural communities, promote infrastructure access and safe housing for the urban poor, and combat social marginalization and extreme poverty so that forced displacement, as well as involuntarily immobility, can be reduced to a minimum and safe mobility can be a successful response and risk reduction strategy.

## Methods
### Vulnerability to flood-induced displacement
We estimate flood-displacement vulnerability as the ratio of displacement to exposure for individual events. First, we geocode the IDMC's Global Internal Displacement Database (GIDD) to assign sub-national administrative units (provinces and districts) to each reported flood-displacement event, where possible. We extract textual location information provided in the GIDD and match it to the names of administrative units in GADM[65]. We are able to identify the location of 1702 out of 3083 recorded displacement events between 2008 and 2018 at a subnational level; the remaining events are assigned the respective country as location. Note that this varying accuracy across the sample may affect the accuracy of the matching between displacement and flood events: Displacement events with available subnational location may be matched more accurately to flood events than displacement events for which only the country is given. Consequently, matching errors may be more likely in earlier records (2008–2012) that have no subnational location information in IDMC. However, we find no systematic difference between vulnerability factors calculated for the full record and for the period of 2013–2018 (Supplementary Fig. S2).

Next, we use the location and timestamp of each displacement event to find associated flood events from the Global Flood Database[33] (GFD), accounting for the possibility of multiple displacement events associated with a single flood event and vice versa[66]. This yields a dataset (FLODIS[66]) of flood extent and the associated number of displacements for 303 events that occurred in 72 countries between 2008 and 2018. This is fewer than the 461 flood events contained in the GFD for that period because for some flood events no corresponding displacement event could be identified. Finally, we multiply flood extent with gridded population data from the Global Human Settlement Layer[67] to estimate exposure, and calculate vulnerability (Supplementary Fig. S17). We also compute for each of the 72 countries the national median vulnerability across all events.

Many previous studies of vulnerability related to damages, mortality, and displacement, particularly at a global scale, have used model-based flood extent estimates[21,27,28,68], which allows studying a greater number of events. However, the accuracy of global flood models remains limited due to uncertainties concerning flood frequency analyses[69], the representation of flood defenses[70], or the river channel geometries[71], and it is unclear whether such models represent individual flood events, and specifically those that have triggered displacement, truthfully enough to be useful for estimating displacement vulnerability. We use remote sensing-based flood extent estimates which represent actual flood events[29] and are likely to yield more realistic estimates of exposure and vulnerability than model-based products. We note that the GFD, and therefore the scope of our study, is limited to fluvial and coastal floods of sufficient size and duration that can be captured by satellite instruments. In particular, these may miss flash floods, as well as small-scale features such as flooded streets in urban areas, or flooding below dense tree cover[33].

**Table 2 | Event-level predictors**

| Predictor | Spatial resolution | Source | Motivation/potential mechanism |
|---|---|---|---|
| Population density (people per km²) | 0.5 arcminute | GHS-Pop[67] | Increased vulnerability of rural areas[22,40]; challenges for evacuation and relief in very high-density urban areas; informal or low-income settlements[24]. |
| Population ≤14 years (% of total population) | 0.5 arcminute | CIESIN[82] | Children lack coping mechanisms and resources during floods, and lack awareness and preparedness before floods[24,51]. |
| Population ≥65 years (% of total population) | 0.5 arcminute | | Physical limitations, place attachment[24,51] (Cutter et al.), (Rufat et al.). "Elderly may have mobility constraints or mobility concerns increasing the burden of care and lack of resilience" (Cutter et al.)[51]. Share of elderly population important predictor of flood-induced fatalities[73]. |
| GDP per capita PPP (2017 international $) | 2nd subnational level (municipality) | Kummu et al.[53] | Wealthier households have higher coping capacity[24]. |
| Urban area (urban area fraction) | 30 arcminute | https://www.isimip.org; based on Goldewijk[83] | "Rural residents may be more vulnerable due to lower incomes and more dependent on locally based resource extraction economies (e.g., farming, fishing). High-density areas (urban) complicate evacuation out of harm's way" (Cutter et al.)[51]. "Especially in the developing burgeoning metropolises, rapid urbanization and population growth are associated with the unregulated sprawl, often with informal settlements and weak infrastructural and economic bases" (Rufat et al., 2015)[24]. |
| Flood protection (return periods) | 1st subnational level (province) | FLOPROS[45] | Higher flood protection standards may be associated with stronger flood emergency response capacities. |
| Event duration (days) | per event | Global Flood Database[33] | Longer floods may cause more damage; residents may be able to weather through shorter floods without having to leave. |
| Total critical infrastructure (-) | 6 arcminute | CISI[84] (composite index including transportation, energy, telecommunication, waste, water, education and health infrastructure) | Access to freshwater supply, healthcare, transportation, energy, and telecommunication infrastructure may increase resilience and facilitate emergency responses. High intensity of critical infrastructure may also indicate good water and wastewater infrastructure (though this is only partly reflected in CISI), which may lower flood hazard and flood damages. |
| Elevation (m) | 0.54 arcminute | Amatulli et al.[85] | Both the physical characteristics of a flood and the form and spatial distribution of assets, such as residential buildings, may differ between plains and mountainous regions. |
| Slope (-) | 0.54 arcminute | | |

## Predictors of vulnerability

In the next step, we identify possible drivers of the vulnerability to flood-induced displacement, drawing on literature related to flood vulnerability and disaster vulnerability more generally[22–24,40,51,72,73]. We selected indicators for which empirical evidence or theoretical arguments suggest a plausible association with vulnerability, and for which data were available for the relevant time period at national or subnational level (Tables 2 and 3). For instance, studying socio-economic covariates of flood-induced fatalities, Reimann et al.[73] found education levels the most significant variable, with the share of elderly population, income inequality, healthcare infrastructure, and rural settlements also influential. On the national level, Toya and Skidmore[74] found economic losses from natural disasters are lower in countries with higher income and higher educational attainment.

We log$_{10}$-transform indicators whose values are very unevenly distributed, such that both our target variable – log$_{10}$(vulnerability) - as well as our predictor variables are either approximately normally or uniformly distributed. Due to the log-transformation, six observations containing zeroes (1 in population density, 3 in urban area, 2 in elevation) were dropped in the event-level analysis, so that the sample there includes only 297 events instead of 303. In the country-level analysis, all 311 events enter the calculation of the national median vulnerability values. We also introduce a dummy variable as a benchmark to measure the significance of a given predictor's importance. The dummy variable is constructed by assigning a random value between 0 and 1 to this predictor for each entry. Descriptive statistics for all indicators can be found in Supplementary Table S3.

## Regression models

Random forests are a tree-based type of supervised machine learning method for classification and regression tasks[37]. Random forest models require no prior assumptions about the functional form of the relationship between predictor and target variable, nor about the existence or absence of interactions between different predictors; and they can handle relatively small datasets[59]. We use the random forest regressor of the Python module "Scikit-learn"[75] version 0.24.1 with the default settings of 100 trees and without sample bootstrapping; no hyperparameter tuning was performed. In the event-level analysis, we account for unobserved country-specific characteristics (Supplementary Fig. S2) by applying a mixed-effects random forest[26] (MERF), with the ISO3 country code as random effect covariate, using the "MERF" Python package version 1.0 (https://github.com/manifoldai/merf).

We construct random forest models using up to three predictors. We test all possible predictor combinations, except for a number of predictors in the country-level analysis which are mutually related and highly correlated (Supplementary Fig. S10) and which we therefore do not combine with one another (Table 3). That is, we do not combine any two of the indicators infant mortality rate, population ≤14 years, population ≥65 years, and population growth, as these are all related to

**Table 3 | Country-level predictors**

| Predictor | Source | Motivation/potential mechanism |
|---|---|---|
| Population density (people per km²) | World Development Indicators[86] | High population density may be associated with advanced flood protection, warning, and response infrastructure |
| *Population ≤14 years (% of total population) | | Human development, poverty; countries with young demographics can be associated with lower human development and resilience to flood risk[24,50,51] |
| *Population ≥65 years (% of total population) | | Human development; high life expectancy indicates well developed health care systems and other social services[47–49]. |
| GDP per capita PPP (current international $) | | Public assets; economic development/wealth can be a proxy for higher resilience against flood risk[24,74] |
| Urban population (% ot total population) | | Advanced infrastructure and services, preparedness and response in urban areas vs. increased vulnerability of rural areas[22,40] |
| **Urban population growth (annual %) | | Rapid growth is associated with human/economic development/informal settlements[24]. |
| *,**Population growth (annual %) | | Rapid growth is associated with informal settlements in flood plains at high risk[24]. |
| *Infant mortality rate (deaths within first year per 1000 live births) | | Human development, poverty, access to healthcare and social services |
| Education Index (average of mean years of schooling and expected years of schooling) | Smits & Permanyer [87] | Economic impacts of disasters lower in countries with higher educational attainment[74]. Education important predictor of flood-induced fatalities[73]. |
| Electoral Democracy Index | V-Dem Dataset[88] | Accountable institutions are associated with more effective protection and response to disasters, but also lower vulnerability to risks associated with floods[18,89,90]. |

* and **, respectively, indicate predictors that are not combined with each other in the random forest models

the age structure of the population and have Pearson correlation coefficients of 0.67 or higher between them; and we do not combine population growth and urban population growth, as the urban population growth rate may closely follow the total population growth rate and the Pearson correlation coefficient between the two variables is 0.9.

### Predictor ranking

As we are interested in which predictors explain vulnerability best, we assess their contribution to explanatory skill across all models (predictor combinations), separately for the event-level and country-level analysis. We first split the data into training sets and test sets that are used to fit the model and predict the vulnerability, respectively. The number of observations is limited, so we choose a leave-one-out cross-validation (LOOCV)[76], splitting the data $n$ times into a single-item test set and a training set of size $n-1$, where $n$ is the total number of observations. To test the capability of explaining variations in vulnerability using the predictor variables, we collect the $n$ vulnerability predictions and compute a single $R^2$ (the coefficient of determination) from the $n$ prediction-observation pairs.

Predictor importance in multivariate models may be obscured by that of other predictors present in a given model. Hence, ranking the importance of variables simply based on $R^2$ values may be misleading. Therefore, we assess predictor importance by measuring the decrease in a model's $R^2$ due to randomly permuting the values of the predictor of interest; based on the hypothesis that if a predictor is of zero importance, randomly permuting its values does not alter the accuracy of the prediction[37]. For each predictor, we calculate the decrease in $R^2$ due to randomization, for all models that include that predictor and have an absolute $R^2 \geq 0.01$ (to exclude models which have a negative $R^2$ to start with). We then rank predictors by the median decrease in $R^2$ across all relevant models.

Alternatively, we also rank predictors by the median increase, due to predictor randomization, in the value of the Akaike (AIC) or Bayesian Information Criterion (BIC). AIC and BIC account for different numbers of predictors between models, and lower values correspond to higher-quality models. While the LOOCV also penalizes overfitting (and shows asymptotic equivalence to these prediction error estimators[77,78]), it does so differently than AIC and BIC, therefore these measures provide complementary ways of model selection or ranking.

### Partial dependence plots

We use partial dependence plots (PDPs)[79] to visualize and analyze the marginal effect of each predictor on the predicted vulnerability. The PDPs are limited to a maximum of two variables, but a collection of PDPs can serve to show the partial dependence of each predictor[80]. We train a random forest model (a MERF model in the event-level analysis) on all data, including the predictors of the model that, in the predictor importance analysis described above, exhibits the highest decrease in $R^2$ after randomizing the predictor of interest. Next, we duplicate the training set 100 times and assign for all values of the predictor of interest one of 100 increments between its minimum and maximum value (across space and time), as similarly done by Vogel et al.[81] Then, for every increment, the trained models predict 100 times an estimated vulnerability value, resulting in a distribution of predictions at each increment (the conditional response distribution; represented in the figures by its median as well as 5th/95th percentile and 33th/66th percentile). The change of this distribution across all increments indicates the marginal effect on vulnerability of the predictor in question, all else being equal. This last assumption is not necessarily realistic, as the joint distribution of predictor values is likely not fully random or stretched out evenly across the range sampled. Also, since the model used to derive the PDPs does not employ any cross-validation, it is prone to overfitting to small-scale features; and the model depends on data density, such that spare intervals with no data for the relevant predictor end up with nearly-horizontal lines. Thus, we do not assign any meaning to these artificial features, but only interpret broad trends over large ranges with continuous data. Given this caveat, the PDPs can nevertheless indicate the direction and approximate functional form of the predictor's effect.

### Reporting summary

Further information on research design is available in the Nature Portfolio Reporting Summary linked to this article.

## Data availability

Flood data is available from the Global Flood Database at (https://global-flood-database.cloudtostreet.ai). Displacement data is available from IDMC's Global Internal Displacement Database at (https://www.internal-displacement.org/database/), and from the Dartmouth Flood Observatory at (https://floodobservatory.colorado.edu/Archives/index.html).

## Code availability

The Python code used to produce the numbers, tables and figures in this paper is available at (https://doi.org/10.5281/zenodo.16620061).

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

## Acknowledgements

This work received funding from the EU Horizon 2020 program, project number 869395 (HABITABLE), and the Federal Ministry of Education and Research (BMBF), funding code 01LS2001A.

## Author contributions

J.S. and K.F. conceived the study. J.S., B.M., and O.K. designed the study. B.M. processed and analyzed data and performed simulations. J.S. and B.M. wrote the paper. All authors including B.D. contributed to the interpretation of the results.

## Funding

## Competing interests

The authors declare no competing interests.
