## [Transparent Peer Review file · Nature Communications]

Socioeconomic predictors of vulnerability to flood-induced displacement

Corresponding Author: Dr Jacob Schewe

Version 0:

Reviewer comments:

Reviewer #1

(Remarks to the Author)

This study contributes to the global assessment of flood-induced displacement vulnerability, using satellite imagery to estimate flooded exposure. By employing satellite data instead of modeled data, the study provides a more realistic and reliable foundation for risk evaluation. This approach helps accurately identify regions and communities most vulnerable to displacement due to floods. However, the limitations of the Global Flood Database and the reduction in sample size potentially skew vulnerability evaluations, especially between rural and urban areas. For example, the finding that rural underinvestment and low human development contribute to flood-induced displacement vulnerability may be influenced by the nature of the data. Given the Global Flood Database's limited sensitivity to urban flood types, rural vulnerability may be overestimated. The authors should clarify whether these results reflect vulnerability patterns or if they are partly shaped by data limitations. Addressing these issues through further discussions and methodological improvements would greatly strengthen the study and broaden its impact.

Data and Methods

- The IDMC displacement data does not originally include geocode information (except for data from 2023). Instead, the authors have geocoded locations based on the region names provided in the dataset. However, missing or imprecise location data for some events, such as those from 2008 that only include country-level information, introduces data gaps. This geocoding process reduces the sample size from 3,087 to 1,702. The authors should further explain this reduction and discuss how it might affect the representativeness of their results.
 - Integrating flood exposure data from the Global Flood Database reduces the sample size to 311, likely because certain flood types, such as flash floods or urban floods, are recorded in the IDMC dataset but missing from the Global Flood Database. This discrepancy may introduce further bias, and the authors should discuss how this impacts the model's robustness and interpretability.
 - As noted in Tellman et al. (2021), the Global Flood Database has limited sensitivity to certain flood types, including urban floods, flash floods, small water channels, and floods under dense canopy cover. This could lead to an overestimation of vulnerability in rural areas and an underestimation in urban areas. It is important for the authors to consider these characteristics when analyzing flood-induced displacement.
- Tellman, B., Sullivan, J.A., Kuhn, C. et al. Satellite imaging reveals increased proportion of population exposed to floods. *Nature* 596, 80–86 (2021). <https://doi.org/10.1038/s41586-021-03695-w>
- Although the GDP data is provided at 5 arcminute, it often lacks sufficient detail, particularly in regions such as Africa. In these areas, large regions are assigned similar GDP values, raising questions about whether the low contribution of GDP to the model is due to insufficient data detail or a lack of significance. While this data may be appropriate for country-level analysis, its suitability for event-based analysis, especially in African regions, remains questionable. Further discussion on this limitation is recommended.
 - There are concerns about whether the reduced sample size is sufficient to support reliable analysis with a random forest model. Random forest generally benefits from larger datasets to enhance robustness and reduce variance. Given the reduction in sample size to around 300, the authors should discuss whether this sample size meets the requirements for their model and how it may impact the reliability of their results.
 - Additionally, the R^2 values of 0.27–0.32 raise questions about the model's fit, and further discussion is needed on whether this level of fit can be considered adequate.

Results and Discussion

- Elevation is strongly associated with vulnerability in this study, yet the reasons behind this relationship, especially in high

or very low areas, remain insufficiently explained. Further analysis on why elevation may robustly influence vulnerability could strengthen the findings.

- The study finds high vulnerability in sparsely populated areas, which may be influenced by the calculation method (displacement/exposed population). This approach tends to produce higher vulnerability scores in areas with smaller exposed populations, even when the number of displacements is low. A more detailed interpretation or discussion of this limitation would be beneficial to clarify these results.
- The study shows that infant mortality contributes to flood-induced displacement vulnerability, yet lacks sufficient discussion on why this factor influences flood-induced displacement. Possible mechanisms could be explored further.
- While FLOPROS contributes little to vulnerability (which may stem from data limitations, particularly in Africa), another infrastructure (CISI) indicators such as drainage and water systems appear more influential. The variation in contributions among infrastructure variables complicates comprehensive interpretation and raises questions about the influence of data quality and differences between indicators. More consistent explanations or additional analyses could improve interpretation. Additional explanations about the varying contributions of infrastructure indicators (e.g., levees vs. drainage systems) would strengthen the study's policy implications, especially regarding how these indicators mitigate flood risk.

Reviewer #2

(Remarks to the Author)

This is an interesting study on how societal characteristics at both national and local level matter for flood vulnerability. The findings largely confirm conclusions from earlier research, and as such the study's claims to novelty are overstated. But the results (if they are robust to additional sensitivity tests) are important and merit publication. While I appreciate the overall effort and contribution, I have some comments and concerns that the authors should address.

The study focuses on "flood vulnerability", operationalized in the aggregate as the ratio of displaced to exposed persons in each flood event. This approach is not entirely unreasonable, but it merits a deeper and more critical reflection and justification, both at the conceptual and empirical level. Conceptually, what does it mean to be exposed? This study treats exposure in a binary fashion as residence in inundated areas, which entails both an inflation of exposure numbers (and thereby deflation of 'vulnerability') in areas with high-rise settlements but also omission of potential cascading impacts that may trigger displacement beyond the flooded areas. Adding to the complexity, displacement also covers preemptive evacuation - including from areas that didn't get flooded (and whose residents therefore do not count in the estimation of exposure). Displacement thus is partly a function of early warning and evacuation capacities, which are inversely related to vulnerability. What are the implications of these issues for the interpretation of your findings? Are the results robust to using deaths-to-exposed as an alternative and arguably simpler metric of 'vulnerability'? Or using displacement as the outcome, controlling for population exposure? Given enduring concerns about what this study really captures, some validation against alternative approaches, supported by evidence from cases, would strengthen confidence in the results.

As alluded to above, the study overstates its contribution in several places. The claim on line 57 that no study so far has attempted to model flood-displacement vulnerability is either willfully ignorant of earlier research or interprets 'vulnerability' very narrowly. Similarly, the claim on line 450 that this study "for the first time" sheds light on the role of social, economic, and political factors ignores a considerable literature going back to Brooks et al. (Global Environmental Change 2005; not cited in study). The paragraph starting on line 481 gives the impression that this is the first study relying on observed flood data instead of modeled-based estimates, which of course is not true. Please revise.

Relatedly, more care should be exercised when the authors discuss their findings in relation to earlier research, for example in the paragraph beginning on line 387. Studies typically differ not only in terms of data and indicators used but also regarding sample size, spatiotemporal domain, etc. Judging from a quick look, it seems that Vestby et al. (2024 cited in the study) have five times as many observations as this study, so the reported differences in findings (line 402 onwards) could simply be a function of different sample characteristics rather than objective differences in modeling design. Further guidance on how findings should be interpreted in relation to the current state of knowledge would be helpful. Given unknown external validity of the analysis, more care should be exercised also when quantifying particular patterns. For example in the discussion on GDP per capita from line 410, which signals undue precision of the shape of the causal relationship (no confidence intervals are given).

The study relies on GFD flood data, which are joined with displacement estimates from IDMC. More information is needed on why the authors elected this approach over using displacement estimates from Dartmouth Flood Observatory, which would have enabled studying the full sample of GFD events. Also, a comparison of displacement estimates between these sources would be appropriate. How well do they match up and what might be reasons for noticeable discrepancies? Can the main results be reproduced if the study uses DFO displacement estimates instead, all else equal?

Other comments:

Why does the study exclude flood events with zero displacement (line 475)? Assuming non-zero exposure, these events contain theoretically relevant information whose omission might introduce bias.

Why does the study use separate models for local and country-level drivers of displacement?

Some of the country-level variables are weakly motivated; demographic characteristics can vary a lot within countries so how can national averages in population density, age distribution etc. influence local flood displacement?

Supplementary Information: Please add descriptive statistics for all indicators.

Figure S11: In the vulnerability box, “affected” should be “exposed” to maintain consistent terminology.

Version 1:

Reviewer comments:

Reviewer #1

(Remarks to the Author)

Overall comments

I would like to thank the authors for their substantial effort in revising the manuscript. The use of updated GDP per capita data, the expanded discussion on elevation and infant mortality, and the clearer statement that the study focuses on fluvial and coastal floods are all significant improvements and are highly appreciated.

I recognize the limitations of the data on flood-induced displacement and appreciate that the authors have worked to extract meaningful insights despite these constraints. At the same time, I encourage the authors to frame their conclusions with appropriate caution, reflecting the uncertainties and the specific scope of the phenomena captured in the analysis.

Therefore, I would also like to propose specific textual revisions, as outlined below.

Specific comments

Title

I apologize for not raising this point during the first round of review, but it remains important. In the title and several parts of the manuscript, the terms “rural underinvestment,” “low human development,” and “low income levels” are used to describe the findings. However, these terms do not directly correspond to the variables analyzed — namely, population density, infant mortality rate, and GDP per capita — and may give readers the impression that development, investment or household income levels were explicitly measured. To avoid misunderstanding, I recommend revising these phrases to more accurately reflect the observed predictors.

On the explanation of R^2 and the interpretation of predictor importance

The manuscript acknowledges the relatively low R^2 values, attributing them to the complexity of the phenomenon and data limitations (e.g., Discussion: “Similar to other studies, a lot of residual variation in vulnerability remains that we cannot explain with our models, with R^2 values rarely exceeding 0.3”). This explanation is reasonable, and I agree that high R^2 values are difficult to achieve in this context.

However, the current phrasing may give the impression that the R^2 values are fully satisfactory and that the models explain vulnerability sufficiently. I recommend rephrasing these sections to more clearly acknowledge the modest explanatory power while still emphasizing that the observed patterns remain meaningful.

In particular, statements such as Lines 366–369 could be softened to reflect that these predictors are relatively more important within the limited explanatory capacity of the models. I also suggest reviewing the conclusion, abstract, and discussion sections to ensure consistency in tone.

Rural–urban comparison

The manuscript emphasizes that vulnerability is higher in rural areas than in urban areas. However, since the analysis focuses on fluvial and coastal floods, with pluvial (urban) floods largely outside its scope, the conclusions about rural–urban differences should explicitly reflect this focus.

In several places (e.g., Lines 216–234), the discussion of the rural–urban context seems to describe general disaster risk, rather than specifically fluvial and coastal floods. While the explanations provided are reasonable and informative, clarifying that these findings pertain to fluvial and coastal floods would avoid potential overgeneralization.

In addition, Lines 212–214 state:

“However, such a bias would imply that we overestimate vulnerability in urban areas, and thus our finding of higher vulnerability in rural areas compared to urban areas remains robust.”

I recommend revising this sentence to explicitly indicate that the finding applies to fluvial and coastal floods.

Terminology of income

GDP per capita is not a measure of household or individual income, and using the term “income” — especially at event-based or sub-national levels — may be misleading.

For example, in the Discussion section:

- Lines 184–186: “Our results show that the variance in flood-displacement vulnerability is better explained by factors other than average income levels alone (Fig. 2, top).”

- Lines 289–292: “The result that GDP per capita is not the most important predictor at sub-national level either suggests that, for similar reasons, income levels — at least as measured with available data products — may inappropriately capture vulnerability even when averaged over smaller areas.”

In both cases, “income levels” may be misinterpreted as referring to individual-level income rather than the aggregate GDP per capita used in the analysis. I recommend clarifying this by explicitly stating, for example, “aggregate income levels as measured by GDP per capita” or another related phrase, to avoid ambiguity.

As noted above, the title also uses “income levels”, which would also need to be rephrased.

Reviewer #2

(Remarks to the Author)

I have read the revised flood study and the revision memo with considerable interest. I believe the authors have satisfactorily addressed most of the issues raised by the reviewers in the previous round. The study has a clear if perhaps narrow contribution to the literature and in my mind warrants publication, pending responding to a few minor issues.

I still find the presentation of the study's novelty in the introduction overstated. Unlike what a reader would infer from reading lines 85 onwards, this study is not the first to use (a) remotely sensed flood data, (b) subnational displacement data, or (c) multivariate models. This concern was raised also in the initial round so I am surprised that the revision hasn't added clarity here. I encourage the authors to rather play up their contribution of using the best available data on observed floods at global scale (GFD) *in combination with* [presumably] the best available subnational displacement estimates (IDMC). This approach is unique and valuable, even if it comes at a cost of having a smaller empirical sample than previous global studies.

Table 1 and Fig. 2: It would be good to also indicate direction of effects for the predictors by adding signs, arrows, or different colors for positive vs. negative effects.

Line 236: My interpretation of the elevation effect is opposite: the marginal effect is highest in flat terrain (floodplains?), drops markedly as soon as some topography is introduced, and thereafter grows only slowly with degree of ruggedness. Consider revising.

Discussion section (lines 165-350): the write-up is rather technical and repetitive (dare I say boring?) and disconnected from the real world. Some references to illustrative cases along the way would help substantiating the analysis. More could be done also in terms of exemplifying the estimated effects: how much would Y change for a given change in X , all else constant?

More could be said about the modest influence of the FLOPROS predictor, which intuitively should have a powerful effect – if current flood protection measures are effective in reducing displacement. The discussion section (e.g., lines 391-404) mentions the importance of physical protection measures, but fails to connect this discussion with the FLOPROS estimate.

REVIEWER COMMENTS

Reviewer #1 (Remarks to the Author):

This study contributes to the global assessment of flood-induced displacement vulnerability, using satellite imagery to estimate flooded exposure. By employing satellite data instead of modeled data, the study provides a more realistic and reliable foundation for risk evaluation. This approach helps accurately identify regions and communities most vulnerable to displacement due to floods. However, the limitations of the Global Flood Database and the reduction in sample size potentially skew vulnerability evaluations, especially between rural and urban areas. For example, the finding that rural underinvestment and low human development contribute to flood-induced displacement vulnerability may be influenced by the nature of the data. Given the Global Flood Database's limited sensitivity to urban flood types, rural vulnerability may be overestimated. The authors should clarify whether these results reflect vulnerability patterns or if they are partly shaped by data limitations. Addressing these issues through further discussions and methodological improvements would greatly strengthen the study and broaden its impact.

Thank you for your encouraging evaluation and your constructive feedback! Below, we respond in detail to each of the points raised.

Data and Methods

- The IDMC displacement data does not originally include geocode information (except for data from 2023). Instead, the authors have geocoded locations based on the region names provided in the dataset. However, missing or imprecise location data for some events, such as those from 2008 that only include country-level information, introduces data gaps. This geocoding process reduces the sample size from 3,087 to 1,702. The authors should further explain this reduction and discuss how it might affect the representativeness of their results.

Thank you for this comment. Indeed, only for 1702 out of 3083 recorded displacement events (between 2008 and 2018) could we identify the location at a subnational level. However, this does not mean that only these events are retained for matching with recorded flood events in the GFD. Instead, we also consider those displacement events for which only the country is available as location information. We then try to match each of these to one or several GFD flood events based on the timing; i.e. by comparing the dates recorded by IDMC and GFD respectively. In doing so, we account for the possibility that one country-level displacement event corresponds to multiple flood events, or vice versa, that multiple displacement events associated with a single flood were separately recorded without providing the subnational location. This procedure is described in Mester et al. (2023), and illustrated in Fig. 2 in that paper (specifically, steps 3 and 4 in the flowchart).

Thus, there is no reduction in sample size due to the geocoding process; rather, the geocoding process aims to achieve the highest possible precision in terms of location of displacement events. It assigns each of the 3083 displacement events a location either at national, provincial (admin 1) or district (admin 2) level. This varying precision across the sample may affect the matching between displacement and flood events; i.e. displacement events for which subnational location information is available may be matched more accurately to flood events than displacement events for which only the country is given. Where there were two different floods in the country in the same year, there is a higher chance to associate the displacement with the “wrong” flood: if, for instance, that flood is recorded to have occurred closer to the date of displacement than the other, “correct” flood. Such errors are less likely when displacement events are more accurately located.

Consequently, matching errors are more likely for early records (2008-2012) for which no subnational location information is provided by IDMC. This is why, in Supplementary Fig. S2, we show calculated vulnerability factors across countries both for the full record and for the period of 2013-2018 only. The comparison of these two cases indicates no systematic difference.

We understand that this information needs to be better conveyed in our submission, and have amended the corresponding explanation in the Methods section (new text underlined):

“We estimate flood-displacement vulnerability as the ratio of displacement to exposure for individual events. First, we geocode the IDMC’s Global Internal Displacement Database (GIDD) to assign sub-national administrative units (provinces and districts) to each reported flood-displacement event, where possible. We extract textual location information provided in the GIDD and match it to the names of administrative units in GADM (GADM, 2018). We are able to identify the location of 1702 out of 3083 recorded displacement events between 2008 and 2018 at a subnational level; the remaining events are assigned the respective country as location. Note that this varying accuracy across the sample may affect the accuracy of the matching between displacement and flood events: Displacement events with available subnational location may be matched more accurately to flood events than displacement events for which only the country is given. Consequently, matching errors may be more likely in earlier records (2008-2012) that have no subnational location information in IDMC. However, we find no systematic difference between vulnerability factors calculated for the full record and for the period of 2013-2018 (Supplementary Fig. S2).

Next, we use the location and timestamp of each displacement event to find associated flood events from the Global Flood Database³² (GFD), accounting for the possibility of multiple displacement events associated with a single flood event and vice versa⁵⁷. This yields a dataset (FLODIS⁵⁷) of flood extent and the associated number of displacements for 303 events that occurred in 72 countries between 2008 and 2018. This is fewer than the 461 flood events contained in the GFD for that period because for some flood events no corresponding displacement event could be identified. Finally, we multiply flood extent with gridded population data from the Global Human Settlement Layer⁵⁸ to estimate

exposure, and calculate vulnerability (Supplementary Fig. S17). We also compute for each of the 72 countries the national median vulnerability across all events.”

We have also revisited the event matching algorithm, and found a small number of events where the merging step had failed. Specifically, multiple IDMC entries associated with the same GFD event had not been merged due to a spelling mistake. After correcting this error, we have a total of 303 events (instead of 311). This correction in the sample marginally improves the model fit: The highest R^2 value achieved in the event-level analysis increases from 0.31 to 0.32, and in the country-level analysis, from 0.32 to 0.34 (see updated Table 1 in our revised manuscript). Likewise, the AIC and BIC values are also somewhat lower now (see updated Tables S1 and S2). The predictor importance ranking, in terms of decrease in R^2 , changes slightly (see updated Fig. 2). Most notably in the event-level analysis, critical infrastructure is now of lower importance, in a similar range as all other predictors except population density and elevation, which have a higher decrease in R^2 (note that in the event-level analysis, also GDP per capita has been replaced with a new dataset, as discussed below, but this has not substantially changed the decrease in R^2 produced when removing this predictor). This has allowed us to simplify the discussion of the results, focusing on these two most important predictors. Similarly, the ranking in the country-level analysis changes slightly, with the percentage of urban population now ranking highest, followed by infant mortality rate. We have adjusted our discussion accordingly; however, the overall results and conclusions do not change.

- Integrating flood exposure data from the Global Flood Database reduces the sample size to 311, likely because certain flood types, such as flash floods or urban floods, are recorded in the IDMC dataset but missing from the Global Flood Database. This discrepancy may introduce further bias, and the authors should discuss how this impacts the model’s robustness and interpretability.

Indeed, sample size is not reduced by the geocoding process, but it is reduced by the matching procedure with GFD flood data. The GFD includes 913 flood events covering the period 2000-2018. Since the IDMC record starts in 2008, 461 out of 913 GFD flood events are already excluded because they occurred earlier. Out of the remaining 461 flood events, 303 could be matched with one or several displacement events. Fig. 1 (top) in Mester et al. (2023) shows the locations of GFD flood events for which no displacement event could be identified (because no displacement occurred, or it was not recorded, or our matching procedure failed), and the locations of displacement events for which no flood event could be identified (e.g. because the flood was not observed or otherwise not included in the GFD, or our matching procedure failed); along with the locations of successful matches. All of these categories are spread somewhat evenly across the globe, indicating that there may not be a strong regional bias introduced by the matching procedure and resulting reduction in sample size. Also note that, because we start with 461 flood events that could potentially be matched with IDMC displacement records, the reduction in sample size due to the matching procedure is not that large (from 461 to 303, i.e. ca. 30% reduction).

Concerning flood types, IDMC records displacement events due to “Flood” hazard, with no further identification of different flood types. As you rightly point out, the GFD generally misses certain flood types such as flash floods, and therefore any displacement events associated with such types of flood are excluded from our final sample. This means our results apply only to the types of floods recorded in the GFD, namely fluvial and coastal floods of sufficient size and duration.

In addition to the changes quoted above, we have clarified the scope of the flood data in the Introduction:

“We combine reported displacement data with remote-sensing data of flood extents and gridded population estimates, to estimate vulnerability, as the ratio between displacement and flood exposure, for over 300 large fluvial and coastal flood events that occurred around the world between 2008 and 2018.”

And in the first part of the Methods section:

“We note that the GFD, and therefore the scope of our study, is limited to fluvial and coastal floods of sufficient size and duration that can be captured by satellite instruments. In particular, these may miss flash floods, as well as small-scale features such as flooded streets in urban areas, or flooding below dense tree cover (Tellman et al., 2021).”

- As noted in Tellman et al. (2021), the Global Flood Database has limited sensitivity to certain flood types, including urban floods, flash floods, small water channels, and floods under dense canopy cover. This could lead to an overestimation of vulnerability in rural areas and an underestimation in urban areas. It is important for the authors to consider these characteristics when analyzing flood-induced displacement.

Tellman, B., Sullivan, J.A., Kuhn, C. et al. Satellite imaging reveals increased proportion of population exposed to floods. *Nature* 596, 80–86 (2021). <https://doi.org/10.1038/s41586-021-03695-w>

Thank you for pointing this out. Indeed, as noted in our manuscript, “An underestimated exposure could [...] arise [...] from incomplete space-borne flood extent observations (Tellman et al., 2021) (for instance, small but important features such as flooded streets in urban areas may not be captured)”. This however would imply that we *overestimate* vulnerability in *urban* areas, since vulnerability is calculated as the ratio of displacement to exposure. Therefore, while it is difficult to ascertain the presence and magnitude of such a bias lacking independent flood extent observations, our qualitative finding of relatively lower vulnerability in urban areas compared to rural areas appears robust to this bias.

We have added the following sentence to our discussion of the event-level results in the Results section:

“We recall that the extent of urban floods may be underestimated e.g. when short-lived or small features such as flash floods or flooded streets are missed by satellite imagery (Tellman et al., 2021); however, such a bias would imply that we overestimate vulnerability in urban areas, and thus our finding of higher vulnerability in rural areas compared to urban areas remains robust.”

- Although the GDP data is provided at 5 arcminute, it often lacks sufficient detail, particularly in regions such as Africa. In these areas, large regions are assigned similar GDP values, raising questions about whether the low contribution of GDP to the model is due to insufficient data detail or a lack of significance. While this data may be appropriate for country-level analysis, its suitability for event-based analysis, especially in African regions, remains questionable. Further discussion on this limitation is recommended.

This is a valid point. In the gridded GDP dataset we used in our original submission (Kummu et al., 2018), GDP per capita was uniform within many countries owing to a lack of subnational GDP data there. The utility of this data as a predictor in the event-level analysis may therefore indeed have been quite limited.

Luckily, an updated dataset has just become available (Kummu et al., 2025) that significantly improves upon the earlier dataset, both by using updated and improved source data and by employing more sophisticated downscaling techniques. While still limited in many low-income countries, the level of geographical detail in this new dataset is higher than in the previous one (compare Fig. 8e in Kummu et al. (2025) to Fig. 2 in Kummu et al. (2018)).

We have replaced the GDP data in our analysis with this new dataset. In combination with the elimination of a few incorrectly matched events in the sample (see above), this has changed the results slightly (see updated Fig. 2, top; reproduced below). Most notably, in the event-level analysis, GDP per capita is now the third most important predictor, while originally it was only the sixth most important predictor. This may be related to the improved quality of the subnational GDP per capita data. At the same time, GDP per capita remains substantially less important than population density and elevation. Thus our conclusions remain unchanged. We have revised the corresponding discussion slightly to reflect the changes, mostly just removing references to critical infrastructure as one of the most important predictors:

“The event-specific analysis indicates population density and elevation as the most important predictors (Fig. 2, top). This result is consistent with the widespread presence of population density and elevation in the models with highest R^2 (Table 1). The ranking also shows that these two predictors are more important than GDP per capita. This finding is crucial, because GDP per capita or some related measure of income levels is often used as the single indicator of socio-economic vulnerability, and assumed to be a reasonable proxy of measures of social status, economic deprivation, etc.^{21,37}. Our

results show that the variance in flood-displacement vulnerability is better explained by factors other than average income levels alone (Fig. 2, top).”

We have also amended the Discussion section to point out the limitations of the subnational GDP data:

“We investigated the importance of a range of social, economic, political, and physical factors in explaining variations in the magnitude of vulnerability across events and across countries. Within countries, using mixed-effects random forest, population density emerges as the most important factor, followed by elevation. At the same time, GDP per capita, a commonly used predictor often associated with disaster risk and vulnerability, is relatively unimportant in our models. We have used the most sophisticated global reconstruction of subnational GDP per capita available(Kummu et al., 2025). This reconstruction utilizes data on urbanization levels, travel time to the closest city, and national-level income inequality to estimate subnational GDP per capita in countries with no reported data, which concerns many African and some Asian countries.”

And we have added a corresponding qualification in the discussion of the country-level results:

“The result that GDP per capita is not the most important predictor at sub-national level either suggests that, for similar reasons, income levels – at least as measured with available data products – may inappropriately capture vulnerability even when averaged over smaller areas.”

- There are concerns about whether the reduced sample size is sufficient to support reliable analysis with a random forest model. Random forest generally benefits from larger datasets to enhance robustness and reduce variance. Given the reduction in sample size to around 300, the authors should discuss whether this sample size meets the requirements for their model and how it may impact the reliability of their results.

We agree with the reviewer that a larger sample would be desirable, and the limited number of events is certainly an important limiting factor for our analysis. That said, random forest models are well suited for small samples (Scornet et al., 2015), and we took care to constrain the number of predictors in our analysis to a small set of variables with plausible relevance for the outcome variable in order to limit the risk of overfitting. Further, we measure out-of-sample predictive skill using leave-one-out cross-validation (LOOCV), thus the reported R^2 values of up to ca. 0.3 mean the sample may be large enough for the model to detect at least some relationships in the data allowing it to extrapolate out of sample. At the same time, these values are still relatively low, and the limited sample size may be one reason.

We have added the following qualification to our Discussion section:

“While random forests are well suited for small samples (Scornet et al., 2015), the limited amount of data available for our analysis introduces considerable variance in our results, which prevents us from drawing more detailed conclusions.”

- Additionally, the R^2 values of 0.27–0.32 raise questions about the model’s fit, and further discussion is needed on whether this level of fit can be considered adequate.

We have extended our related discussion in the Discussion section as follows:

“While random forests are well suited for small samples (Scornet et al., 2015), the limited amount of data available for our analysis introduces considerable variance in our results, which prevents us from drawing more detailed conclusions. Similar to other studies, a lot of residual variation in vulnerability remains that we cannot explain with our models, with R^2 values rarely exceeding 0.3. This variance may be partly related to data quality issues regarding the measurement of displacement. While IDMC to our knowledge is the highest-quality global source of displacement data, no uncertainty estimates are available for their figures, and IDMC figures and those from another global data provider (the Dartmouth Flood Observatory, DFO) can differ by several orders of magnitude (Supplementary Fig. S11). That said, some of our main findings – the prime importance of population density in the event-level analysis; the role of infant mortality rate, urban population, and share of elderly population among the most important predictors in the country-level analysis; and the relatively lower importance of GDP per capita – are robust even when DFO displacement data are used instead of IDMC data

(Supplementary Fig. S12 and S13; note R² values are lower with DFO data than with IDMC data, Supplementary Fig. S14-S16).

Results and Discussion

- Elevation is strongly associated with vulnerability in this study, yet the reasons behind this relationship, especially in high or very low areas, remain insufficiently explained. Further analysis on why elevation may robustly influence vulnerability could strengthen the findings.

Thank you for this suggestion. We have added the following discussion of potential reasons behind the role of elevation:

“The marginal effect of the second most important predictor, elevation, is largely positive, such that vulnerability increases with elevation. Floods in mountainous regions tend to have different properties than floods in low-lying areas, for example mainly higher velocity of flow, or potentially damaging debris carried by the water (Fuchs et al., 2019; Jakob et al., 2022). At the same time, mountain regions often have a different socioeconomic structure than lowlands: infrastructure and economic development are heavily modulated by topography; and a common demographic pattern is that young people move to cities while older people remain in mountain villages and towns (Frigerio et al., 2016; Sung & Liaw, 2021). The increased vulnerability at higher elevations may thus partly reflect differences in age, educational, and economic characteristics influencing vulnerability and adaptive capacity.”

And in the Discussion section:

“Our finding that vulnerability tends to be higher in high-elevation and very low-elevation areas may be related to physical characteristics of mountain floods, but also socioeconomic and demographic differences between mountain regions and lowlands.”

- The study finds high vulnerability in sparsely populated areas, which may be influenced by the calculation method (displacement/exposed population). This approach tends to produce higher vulnerability scores in areas with smaller exposed populations, even when the number of displacements is low. A more detailed interpretation or discussion of this limitation would be beneficial to clarify these results.

Thank you for the opportunity to clarify our approach. Indeed, by calculating vulnerability as the ratio of displacement to exposed population, we get higher vulnerability scores in areas with smaller exposed populations, given the same number of displacements. This design is on purpose: It is expected that the size of the exposed population explains some of the variation in displacement, but we are interested in what might explain the remaining variation, after controlling for exposure. Fig. S1 shows that vulnerability varies nearly as much as does displacement itself, and this variation is observed across the spectrum from small (hundreds of displacements) to very large (millions of displacements) events. Thus, even if one regards small events as less important (though mind that many small events can add up to large displacement

numbers), high vulnerability scores in our analysis are not only contributed by small events but also by large events. At the same time, the smallest events in our sample - likely occurring in sparsely populated areas - are associated with low vulnerability scores (Fig. S1 bottom). Thus, the finding of high vulnerability in sparsely populated areas is not an artefact of our calculation method.

We have extended our interpretation in the Discussion section to clarify this aspect:

“While sparsely populated, rural areas have been shown to be especially vulnerable to flood-related impacts in empirical and qualitative studies based on data from the USA (Cutter et al., 2016) and across the world (Cross, 2001; Jamshed et al., 2020; Rufat et al., 2015), this finding is new in the context of displacement. This result underlines the importance of clearly separating the effect of population density on vulnerability from its role in determining exposure: in densely populated areas, more people will be exposed to a given flood event. Studies trying to predict displacement using population density as an explanatory variable may conflate the two effects and produce biased results. In our study we predict (the log of) displacement *vulnerability*, as the ratio between displacement and exposure, thus accounting for effects of population density on exposure. The remaining effect of population density on vulnerability, highlighted by our models, indicates residents of rural areas on average suffer higher displacement risk than their urban counterparts, for a given hazard.”

- The study shows that infant mortality contributes to flood-induced displacement vulnerability, yet lacks sufficient discussion on why this factor influences flood-induced displacement. Possible mechanisms could be explored further.

Thank you for this comment. We have extended our discussion on the possible mechanisms related to the infant mortality rate indicator, in the Discussion section:

“Across countries, the most important factors explaining differences in the average order of magnitude of vulnerability, according to random forest models, are urbanization and infant mortality rate, followed by the share of elderly population and GDP per capita. When model quality is measured by AIC or BIC instead of R², education is similarly ranked very highly. These observations are in line with general theories stating that a higher level of human development leads to a decrease in vulnerability^{5,52}. Age structure, health, infrastructure, and education indicators are often found indicative of vulnerability to climate extremes for communities and individuals⁵³. Infant mortality rate may serve as a proxy for public service delivery in the context of climate extremes, such as emergency and recovery assistance, and have been found to explain parts of the variation in disaster-related deaths also within countries⁵⁴. High infant mortality rates are also related to the marginalization of affected communities⁵⁵, which is associated with higher flood risk⁵⁶, caused by, for example, a lack of early warning systems, physical protection measures, or official emergency and recovery support. More directly, infant mortality can

contribute to displacement during and after floods by acting as both a health crisis and a socio-emotional tipping point for affected households, compelling them to flee high-risk zones. In regions where floodwaters damage healthcare infrastructure or impede access to essential services, the elevated risk of infant death due to preventable causes—such as waterborne diseases or lack of neonatal care—can push families to migrate in search of safer environments with better medical access.”

- While FLOPROS contributes little to vulnerability (which may stem from data limitations, particularly in Africa), another infrastructure (CISI) indicators such as drainage and water systems appear more influential. The variation in contributions among infrastructure variables complicates comprehensive interpretation and raises questions about the influence of data quality and differences between indicators. More consistent explanations or additional analyses could improve interpretation. Additional explanations about the varying contributions of infrastructure indicators (e.g., levees vs. drainage systems) would strengthen the study’s policy implications, especially regarding how these indicators mitigate flood risk.

We regret that our description and interpretation of the infrastructure indicator may have been misleading. Indeed, the CISI index measures various infrastructure not only related to water but also to healthcare, transportation, or energy. In terms of water-related infrastructure, it only includes water supply infrastructure such as reservoirs and water works, and does not explicitly include any drainage or flood protection infrastructure. Therefore, it complements the FLOPROS indicator which measures flood protection standards.

We have added a short description of the CISI indicator in Table 2:

“CISI (Nirandjan et al., 2022) (composite index including transportation, energy, telecommunication, waste, water, education and health infrastructure)”

And we have amended our explanation in the “Motivation/potential mechanism” column in Table 2, which was ambiguous before:

“Access to freshwater supply, healthcare, transportation, energy, and telecommunication infrastructure may increase resilience and facilitate emergency responses. High intensity of critical infrastructure may also indicate good water and wastewater infrastructure (though this is only partly reflected in CISI), which may lower flood hazard and flood damages.“

In any case, after correcting the merging of some events in our flood-displacement database, and introducing an improved subnational GDP dataset, as described above, the total critical infrastructure indicator has become somewhat less important in our models, with very similar decrease in R^2 values as FLOPROS. Thus, neither of these indicators emerges as a particularly important determinant of vulnerability from our analysis.

Reviewer #2 (Remarks to the Author):

This is an interesting study on how societal characteristics at both national and local level matter for flood vulnerability. The findings largely confirm conclusions from earlier research, and as such the study's claims to novelty are overstated. But the results (if they are robust to additional sensitivity tests) are important and merit publication. While I appreciate the overall effort and contribution, I have some comments and concerns that the authors should address.

The study focuses on "flood vulnerability", operationalized in the aggregate as the ratio of displaced to exposed persons in each flood event. This approach is not entirely unreasonable, but it merits a deeper and more critical reflection and justification, both at the conceptual and empirical level. Conceptually, what does it mean to be exposed? This study treats exposure in a binary fashion as residence in inundated areas, which entails both an inflation of exposure numbers (and thereby deflation of 'vulnerability') in areas with high-rise settlements but also omission of potential cascading impacts that may trigger displacement beyond the flooded areas. Adding to the complexity, displacement also covers preemptive evacuation - including from areas that didn't get flooded (and whose residents therefore do not count in the estimation of exposure). Displacement thus is partly a function of early warning and evacuation capacities, which are inversely related to vulnerability. What are the implications of these issues for the interpretation of your findings? Are the results robust to using deaths-to-exposed as an alternative and arguably simpler metric of 'vulnerability'? Or using displacement as the outcome, controlling for population exposure? Given enduring concerns about what this study really captures, some validation against alternative approaches, supported by evidence from cases, would strengthen confidence in the results.

Thank you for these important and encouraging comments. You are right that there are ambiguities in delineating vulnerability from exposure. We have highlighted this in our discussion of the role of population density:

"Vulnerability to floods and other disasters can be larger in rural areas than in cities for physical but also social and economic reasons (Cross, 2001). In physical terms, small rural communities may have a much larger share of their population or assets exposed to a given hazard than large cities. In an urban context, much of the population living in the area for which exposure and vulnerability are assessed (e.g. some administrative unit or a grid cell) may be less exposed to hazardous or damaging water levels e.g. because of variations in elevation across the city, and multi-story residential buildings or other infrastructure may provide refuge and prevent displacement. In contrast, a small village may get completely flooded quickly, offering little for its inhabitants to take refuge, and making it much more likely that most or all of its population may be displaced. These physical aspects concern fine-scale variations in exposure, which our data cannot

distinguish, and which are thus subsumed in our vulnerability metric. In terms of social and economic reasons, rural areas tend to be poorer, with lower structural resilience of buildings, and to be neglected or treated subordinately by centralized government. For example, levees that protect larger settlements may lead to even higher flood levels for neighboring or downstream, smaller settlements. Rural areas may also lack economies of scale, as cities can afford much larger emergency response capacities such as professional fire brigades (Cross, 2001; Cutter et al., 2016). Resilience against floods and other disasters differs markedly between rural and urban counties in the USA (Cutter et al., 2016); such differences are likely to be more pronounced in less wealthy countries.”

Your comment about preemptive evacuations is also correct, and we address this problem in our Discussion. In addition, very recently, the IDMC has begun providing disaggregated estimates of disaster-induced displacements, distinguishing between preemptive evacuations and post-disaster displacements (available at <https://www.internal-displacement.org/database/displacement-data/> for the year 2023). This level of detail is not available for the period covered in our study. However, it is worth noting that in 2023, out of 3180 flood-induced displacement events, only 46 were recorded to be associated with preemptive evacuations, while 2171 were associated with post-disaster displacement (963 events have no information on the type of displacement). Most of the events with preemptive evacuations took place in high-income countries, though some occurred in Congo, Algeria, Myanmar, Nigeria, Rwanda, and Tchad. We have included these insights in our discussion:

“Moreover, the available data fail to distinguish post-disaster displacement from preemptive evacuations. While the latter can save lives, both forms of displacement are associated with costs and burden (McAdam, 2022). Our results do not appear to be primarily driven by the presence of preemptive evacuations in the data: sparsely populated areas, identified as particularly vulnerable in our study, are less likely to receive early warnings and to be targeted for planned evacuation than urban centers; and countries with high infant mortality rates are less likely to have the financial and institutional resources required for organizing effective evacuation campaigns. Disaggregated estimates by the Internal Displacement Monitoring Center (IDMC) (Global Internal Displacement Database, n.d.) for 2023 show that out of 3180 flood-induced displacement events in that year, only 46 were associated with preemptive evacuations, while 2171 were associated with post-disaster displacement (963 events had no related information). This record indicates that preemptive evacuations may constitute a minor fraction of all displacement events. Nevertheless, the missing distinction between post-disaster displacement and preemptive evacuations is a major caveat of our study, and may put important limits on the overall explanatory power of our models.”

We have also conducted the alternative analyses you kindly suggested. Using the ratio of deaths-to-exposed, instead of displaced-to-exposed, we still find population density and elevation among the three most important predictors in the event-level models; and infant mortality rate as the second-most important predictor in the country-level models. This suggests

that these indicators may represent general aspects of flood vulnerability not specific to displacement. As another commonality, GDP per capita is not among the most important predictors of the deaths-to-exposed ratio. There are also differences: In the country-level analysis, the educational index is the most important predictor, and urbanization level the least important predictor, of the deaths-to-exposed ratio. This is contrary to the main analysis of displaced-to-exposed ratio, where urbanization level is ranked most important and education is ranked relatively low. However, we have data on the number of fatalities linked to observed floods for only 195 events in 55 countries, and so differences especially in the country-level analysis may partly be due to the smaller sample that, for the ratio of deaths-to-exposed excludes a number of countries present in the main analysis.

We have included the results of this analysis in the Supplement, and refer to them in our Results section:

“The event-specific analysis indicates population density and elevation as the most important predictors (Fig. 2, top). This result is consistent with the widespread presence of population density and elevation in the models with highest R^2 (Table 1). The ranking also shows that these two predictors are more important than GDP per capita. This finding is crucial, because GDP per capita or some related measure of income levels is often used as the single indicator of socio-economic vulnerability, and assumed to be a reasonable proxy of measures of social status, economic deprivation, etc. (Fox et al., 2024; Kakinuma et al., 2020). Our results show that the variance in flood-displacement vulnerability is better explained by factors other than average income levels alone (Fig. 2, top). These results are robust when using the ratio of deaths to exposure as an alternative target variable (Supplementary Fig. S8), suggesting they may represent general aspects of flood vulnerability.”

As for using displacement as the outcome, controlling for population exposure, these models also have population density and infant mortality rate among the two most important predictors (besides exposure itself) in the event-level and country-level analysis, respectively (see Fig. R1 below, top). However, these models all have much lower R^2 values (0.175 at most) than the models using displacement vulnerability as the outcome (figure below, bottom). We believe this supports our approach of including exposure directly in the target variable, i.e. the displaced-to-exposed ratio. Despite the conceptual ambiguities in our definition of exposure, discussed above, our estimate of the number of people exposed to flooding is expected for theoretical reasons (only exposed people can get displaced; bar the caveats discussed above) to have a measurable effect on displacement.

Figure R1: Number of displacements as outcome variable in event-level (left) and country-level analysis (right). Top, decrease in R^2 ; bottom, total R^2 .

As alluded to above, the study overstates its contribution in several places. The claim on line 57 that no study so far has attempted to model flood-displacement vulnerability is either willfully ignorant of earlier research or interprets ‘vulnerability’ very narrowly. Similarly, the claim on line 450 that this study “for the first time” sheds light on the role of social, economic, and political factors ignores a considerable literature going back to Brooks et al. (Global Environmental Change 2005; not cited in study). The paragraph starting on line 481 gives the impression that this is the first study relying on observed flood data instead of modeled-based estimates, which of course is not true. Please revise.

We regret creating a false impression of the contribution of our study. The first claim referred to modelling flood-displacement vulnerability at the global scale, and we are indeed not aware of any other study doing this. However, our wording may have been ambiguous, and the claim is not critical to our arguments, so we have removed it:

“However, it is unclear how displacement vulnerability varies between flood events, and which factors, beyond differences in hazard and exposure, might explain this variation. Only few studies have explored flood-induced displacement at the global scale (Kakinuma et al., 2020; Kam et al., 2021; Ronco et al., 2023; Vestby et al., 2024).”

The second claim, in the Discussion section, was also ambiguous, and we have extended it to clarify the specific contribution of our study:

“Besides this, our study for the first time sheds light on the role of several social, economic, and political factors, for the variability in flood-induced displacement outcomes between as well as within countries. It thus adds to the literature on the determinants of vulnerability to climate events (Brooks et al., 2005).”

And we have extended the related statement in the Introduction, including the reference to Brooks et al. (2005):

“Similarly, little is known about the role of non-economic or local factors, such as urban development and infrastructure access, demographics, or social disparities, which are important drivers of social vulnerability to flooding in many case studies (Cross, 2001; Fatemi et al., 2017; Rufat et al., 2015) and large-scale assessments (Brooks et al., 2005) but have rarely been considered in relation to displacement.”

Finally, we have clarified the statement on using observed flood data instead of modeled-based estimates, in the Methods section:

“Many previous studies of vulnerability related to damages, mortality, and displacement, particularly at a global scale, have used model-based flood extent estimates (Jongman et al., 2015; Kakinuma et al., 2020; Sauer et al., 2021; Tanoue et al., 2016), which allows studying a greater number of events.”

Relatedly, more care should be exercised when the authors discuss their findings in relation to earlier research, for example in the paragraph beginning on line 387. Studies typically differ not only in terms of data and indicators used but also regarding sample size, spatiotemporal domain, etc. Judging from a quick look, it seems that Vestby et al. (2024 cited in the study) have five times as many observations as this study, so the reported differences in findings (line 402 onwards) could simply be a function of different sample characteristics rather than objective differences in modeling design. Further guidance on how findings should be interpreted in relation to the current state of knowledge would be helpful. Given unknown external validity of the analysis, more care should be exercised also when quantifying particular patterns. For example in the discussion on GDP per capita from line 410, which signals undue precision of the shape of the causal relationship (no confidence intervals are given).

We agree that comparison between different studies is complicated due to many methodological differences. In the paragraph in question, we aim to point out some of the major differences and similarities in results between our study and the two most closely related previous studies. We also highlight some major methodological differences which we believe are important both for interpreting the results and for understanding the conceptual advances of our study; however we do not claim that these particular methodological differences exclusively explain the differences in results.

Vestby et al. (2024) indeed have about four times as many observations as our study (note that they drop about a quarter of events due to missing displacement information). This is partly due to their use of displacement data directly from DFO (which is available for more years compared to IDMC data), and partly because they assume multicountry flood events (for each of which only a single displacement number is reported in DFO) trigger displacement in each affected country, proportional to the relative population exposure. We avoid this because it effectively creates new displacement observations that have not been reported. We instead rely only on displacement figures and associated geographies reported directly by the IDMC.

The study relies on GFD flood data, which are joined with displacement estimates from IDMC. More information is needed on why the authors elected this approach over using displacement estimates from Dartmouth Flood Observatory, which would have enabled studying the full sample of GFD events. Also, a comparison of displacement estimates between these sources would be appropriate. How well do they match up and what might be reasons for noticeable discrepancies? Can the main results be reproduced if the study uses DFO displacement estimates instead, all else equal?

We prefer IDMC data for two reasons. The first is the fact that DFO provides only a single number of displacements for each flood event, also when the flood extends across country borders. As mentioned above, this requires additional, strong assumptions when assigning displacements to countries (which we need to do for our country-level analysis, though not necessarily for the event-level analysis). The second reason is that in our view the IDMC data are more reliable. IDMC collects information from different, governmental as well as non-governmental sources, reviews those figures and assesses the quality of each source, to come up with a final estimate of displacements. DFO relies mainly on news reports to produce displacement estimates, sometimes employing rules of thumb to translate categorical estimates into precise numbers (e.g. "If the news report only mentions that "thousands were evacuated", the number is estimated at 3000.", <https://floodobservatory.colorado.edu/Archives/ArchiveNotes.html>).

We acknowledge it is difficult to understand exactly the procedures employed by each organization, and to assess the validity of their estimates, given the lack of any independent ground truth data. The estimates in fact sometimes differ enormously between the two sources, where one may report a few hundred displacements and the other hundreds of thousands (Supplementary Fig. S11; reproduced below). This indicates uncertainties in at least one, and

potentially both, of the datasets. As suggested, we have run our analysis using DFO estimates. A first observation is that our models are less able to explain the variation in vulnerabilities with DFO, compared to IDMC. Even when taking advantage of the larger sample size available with DFO, the highest R^2 values in our event-level and country-level analysis are about 0.22 and 0.30, respectively (Supplementary Fig. S14); compared to 0.31 and 0.34, respectively, using our main IDMC sample (Supplementary Fig. S5). When considering only those events for which both IDMC and DFO estimates are available, the highest R^2 values reduce further to about 0.14 and 0.18 in the event-level and country-level analysis, respectively, for DFO (Supplementary Fig. S15); compared to 0.28 and 0.31, respectively, using IDMC (Supplementary Fig. S16). This may support our assumption of a higher accuracy of the IDMC data; although it is also possible our models are less appropriate than the main analysis suggests.

The second observation is that some of our main findings are insensitive even to replacing the source of displacement data: Using DFO, population density is still the most important predictor (by decrease in R^2) in the event-level analysis; infant mortality rate, urban population, and share of elderly population are among the most important predictors in the country-level analysis; and all these indicators are ranked more important than GDP per capita (Supplementary Fig. S12 and S13). Of course there are also many differences; notably, elevation is ranked much lower in the event-level analysis using DFO than when using IDMC. It should be noted that for this comparison exercise, we also adjusted the IDMC sample such that both DFO and IDMC samples contain only 270 events for which non-zero estimates are available from both sources. Even this trimming of the IDMC sample (from 303 down to 270 events) has induced some changes compared to the main analysis (compare Supplementary Fig. S13 versus Fig. 2 in the main paper). Most notably, and in line with our initial results, the ranking of the education indicator appears to be quite sensitive to changes in the sample, but also to the metric: It is ranked second most important in terms of median increase in AIC or BIC (Supplementary Fig. S6 and S7), and even most important in terms of median decrease in R^2 in the trimmed sample (Supplementary Fig. S13), but only sixth most important in terms of median decrease in R^2 in the main sample (Fig. 2; though note that even there, this predictor contributes quite a lot in specific, individual models, shown by the extent of the right-hand-side whisker).

We now explain this issue and our additional analysis in the Discussion section:

“While random forests are well suited for small samples⁵⁸, the limited amount of data available for our analysis introduces considerable variance in our results, which prevents us from drawing more detailed conclusions. Similar to other studies, a lot of residual variation in vulnerability remains that we cannot explain with our models, with R^2 values rarely exceeding 0.3. This variance may be partly related to data quality issues regarding the measurement of displacement. While IDMC to our knowledge is the highest-quality global source of displacement data, no uncertainty estimates are available for their figures, and IDMC figures and those from another global data provider (the Dartmouth Flood Observatory, DFO) can differ by several orders of magnitude (Supplementary Fig. S11). That said, some of our main findings – the prime importance of population density in the event-level analysis; the role of infant mortality rate, urban population, and share

of elderly population among the most important predictors in the country-level analysis; and the relatively lower importance of GDP per capita – are robust even when DFO displacement data are used instead of IDMC data (Supplementary Fig. S12 and S13; note \$R^2\$ values are lower with DFO data than with IDMC data, Supplementary Fig. S14-S16).”

Figure S11. Comparison of displacement data from DFO and IDMC, for 335 flood events with non-zero displacements according to at least one of the sources.

Other comments:

Why does the study exclude flood events with zero displacement (line 475)? Assuming non-zero exposure, these events contain theoretically relevant information whose omission might introduce bias.

We exclude flood events for which no corresponding displacement event could be identified. It is not clear whether these events actually had zero displacement (“true” zero) or are missing information about displacements. For DFO, Vestby et al. (2024) exclude events with zero displacements based on the assumption that these reflect missing data. Indeed, supporting this assumption, their Fig. 5 shows a set of data points with zero displacements clearly separated from the rest of the data. For IDMC data, too, we interpret absence of a displacement record as missing information, rather than as true zero.

Why does the study use separate models for local and country-level drivers of displacement?

Thank you for the opportunity to clarify our rationale. We believe that the drivers of displacement, and specifically also the determinants of vulnerability, ultimately are local. Therefore we use geolocated displacement data in conjunction with spatially accurate flood observations to determine the exact locations of flood-exposed populations associated with reported displacement, along with the corresponding local socioeconomic and physical conditions potentially related to vulnerability. This high level of spatial detail is one of the main advances of our study over previous work.

At the same time, not all possible determinants of vulnerability can be directly measured currently, especially for a large set of events across many countries. For example, the presence of disaster early warning systems, local-scale physical protection measures, and the availability of emergency and recovery assistance or public awareness to flood hazards are not known. Such factors may however be associated with national-level characteristics such as public wealth or form of governance, for instance because countries with a high level of government accountability and sufficient public budget may have relatively good emergency infrastructure in place across locations. Also, some indicators available only at the national level, such as infant mortality rate, may be good proxies for some of the unmeasured local characteristics driving vulnerability. Therefore, country-level indicators may be able to explain some more of the variation in vulnerability across countries than can be explained solely with the indicators available at the local level, and provide additional insight into the determinants of vulnerability.

Following this rationale, we model vulnerability at the event level using mixed-effects random forest (MERF) because events occurring in one country may systematically differ from events occurring in another country. The MERF accounts for such clustering of the data. In the country-level analysis, on the other hand, we model the median vulnerability of each country using regular random forest, as we do not assume any clustering here.

We have extended our explanation in the Introduction as follows:

“We combine reported displacement data with remote flood extent observations and gridded population estimates, to estimate vulnerability, as the ratio between displacement and flood exposure, for over 300 large fluvial and coastal flood events that occurred around the world between 2008 and 2018. We then examine which predictors, measured at high sub-national resolution, explain most of the observed variation in displacement vulnerability between individual events, using a mixed-effects random forest²⁶ to account for unobserved country-specific factors (i.e. vulnerability might be systematically lower in one country than in another). To gain insight into these potential country-specific factors, we apply random forest regression to predict the median vulnerability per country using predictors measured at the national level. While vulnerability is ultimately relevant at the local level, it is impossible to directly measure all its possible determinants across many countries. Factors such as the presence of

disaster early warning systems, physical protection measures, the availability of emergency and recovery assistance, or public awareness to flood hazards are hardly documented at global resolution. Such elements may however be reflected by national-level characteristics such as public assets or forms of governance. Hence, country-level indicators might explain some of the variation in vulnerability across countries as opposed to indicators only available at the local level.”

Some of the country-level variables are weakly motivated; demographic characteristics can vary a lot within countries so how can national averages in population density, age distribution etc. influence local flood displacement?

As mentioned just above, some of the country-level indicators are proxies of unobserved local characteristics that may be similar across different regions in a country. Specifically, age distribution has been shown to be strongly associated with poverty levels and general levels of human development; and countries with high population density may tend to have advanced flood protection, warning, and response infrastructure in place compared to low-density countries (even though this predictor turns out to be of low importance in our models).

We have improved the description of the motivation for the country-level predictors in Table 2, as follows:

Country-level		
Predictor	Source	Motivation/potential mechanism
Population density (people per km ²)	World Development Indicators ⁷⁸	High population density may be associated with advanced flood protection, warning, and response infrastructure
*Population ≤ 14 years (% of total population)		Human development, poverty; countries with young demographics can be associated with lower human development and resilience to flood risk^{24,49,50}

*Population ≥ 65 years (% of total population)		Human development; high life expectancy indicates well developed health care systems and other social services⁴⁶⁻⁴⁸.
GDP per capita PPP (current international \$)		Public assets; economic development/wealth can be a proxy for higher resilience against flood risk^{24,72}
Urban population (% of total population)		Advanced infrastructure and services, preparedness and response in urban areas vs. increased vulnerability of rural areas^{22,40}
**Urban population growth (annual %)		Rapid growth is associated with human/economic development/informal settlements²⁴.
*, **Population growth (annual %)		Rapid growth is associated with informal settlements in flood plains at high risk²⁴.
*Infant mortality rate (deaths within first year per 1,000 live births)		Human development, poverty, access to healthcare and social services
Education Index (average of mean years of schooling and expected years of schooling)	Smits & Permanyer (2019)⁸⁰	Economic impacts of disasters lower in countries with higher educational attainment⁷². Education important predictor of flood-induced fatalities⁷¹.

Electoral Democracy Index	V-Dem Dataset ⁸¹	Accountable institutions are associated with more effective protection and response to disasters, but also lower vulnerability to risks associated with floods (Ahrens & Rudolph, 2006; Peduzzi et al., 2009; Vestby et al., 2024)
-----------------------------	---

Supplementary Information: Please add descriptive statistics for all indicators.

We have added a table showing descriptive statistics in SI, and refer to it in the Methods section of the main paper:

“Descriptive statistics for all indicators can be found in Supplementary Table S3.”

Figure S11: In the vulnerability box, “affected” should be “exposed” to maintain consistent terminology.

Thank you, we have adjusted the Figure (now Fig. S17) accordingly.

Note:

We have also made small textual improvements throughout the manuscript, visible in the tracked-changes version of the revised manuscript.

REVIEWER COMMENTS

Reviewer #1 (Remarks to the Author):

Overall comments

I would like to thank the authors for their substantial effort in revising the manuscript. The use of updated GDP per capita data, the expanded discussion on elevation and infant mortality, and the clearer statement that the study focuses on fluvial and coastal floods are all significant improvements and are highly appreciated.

I recognize the limitations of the data on flood-induced displacement and appreciate that the authors have worked to extract meaningful insights despite these constraints. At the same time, I encourage the authors to frame their conclusions with appropriate caution, reflecting the uncertainties and the specific scope of the phenomena captured in the analysis.

Therefore, I would also like to propose specific textual revisions, as outlined below.

We are grateful for the evaluation and constructive feedback, and have addressed the comments and suggestions as follows.

Specific comments

Title

I apologize for not raising this point during the first round of review, but it remains important. In the title and several parts of the manuscript, the terms “rural underinvestment,” “low human development,” and “low income levels” are used to describe the findings. However, these terms do not directly correspond to the variables analyzed — namely, population density, infant mortality rate, and GDP per capita — and may give readers the impression that development, investment or household income levels were explicitly measured. To avoid misunderstanding, I recommend revising these phrases to more accurately reflect the observed predictors.

We have changed the title to “Socioeconomic predictors of vulnerability to flood-induced displacement”. This avoids the misunderstanding you allude to, while also avoiding potential misunderstandings of the names of the variables analyzed, as we cannot provide further context in the title.

On the explanation of R^2 and the interpretation of predictor importance

The manuscript acknowledges the relatively low R^2 values, attributing them to the complexity of the phenomenon and data limitations (e.g., Discussion: “Similar to other studies, a lot of residual variation in vulnerability remains that we cannot explain with our models, with R^2 values rarely exceeding 0.3”). This explanation is reasonable, and I agree that high R^2 values are difficult to achieve in this context.

However, the current phrasing may give the impression that the R^2 values are fully satisfactory

and that the models explain vulnerability sufficiently. I recommend rephrasing these sections to more clearly acknowledge the modest explanatory power while still emphasizing that the observed patterns remain meaningful.

In particular, statements such as Lines 366–369 could be softened to reflect that these predictors are relatively more important within the limited explanatory capacity of the models. I also suggest reviewing the conclusion, abstract, and discussion sections to ensure consistency in tone.

We have qualified said statements by explicitly citing the maximum R^2 values achieved with the corresponding models:

“Within countries, using mixed-effects random forest models ($R^2 \leq 0.31$), population density emerges as the most important factor, followed by elevation.”

“Across countries, the most important factors explaining differences in the average order of magnitude of vulnerability, according to random forest models ($R^2 \leq 0.34$), are urbanization and infant mortality rate, followed by the share of elderly population and GDP per capita.”

Moreover, early on in the first part of the results section, we have included the following clarifying statement:

“The modest R^2 values, while not surprising given the complexity of the issue, mean our models only partially explain vulnerability. The predictor importance ranking discussed below must be viewed in this context of low explained variance; nevertheless, they provide meaningful insights into the relative roles of different socioeconomic factors.”

Rural–urban comparison

The manuscript emphasizes that vulnerability is higher in rural areas than in urban areas. However, since the analysis focuses on fluvial and coastal floods, with pluvial (urban) floods largely outside its scope, the conclusions about rural–urban differences should explicitly reflect this focus.

In several places (e.g., Lines 216–234), the discussion of the rural–urban context seems to describe general disaster risk, rather than specifically fluvial and coastal floods. While the explanations provided are reasonable and informative, clarifying that these findings pertain to fluvial and coastal floods would avoid potential overgeneralization.

In this particular paragraph, we put our rural-urban findings into the context of relevant literature. Some of this literature is indeed related to general disaster risk, as indicated at the beginning of the paragraph (“Vulnerability to floods and other disasters can be larger in rural areas than in cities”). We aim to show how our finding – higher vulnerability in rural than in urban areas specifically for displacement induced by fluvial and coastal floods – aligns with and supports the more general finding in previous studies, of higher vulnerability in rural than in urban areas for

other outcomes than displacement, and for floods as well as other disasters. We have amended the paragraph to clarify this further:

“Vulnerability to floods and other disasters can be larger in rural areas than in cities for physical but also social and economic reasons²². In physical terms, small rural communities may have a much larger share of their population or assets exposed to a given hazard than large cities. For fluvial and coastal floods, in an urban context, much of the population living in the area for which exposure and vulnerability are assessed (e.g. some administrative unit or a grid cell) may be less exposed to hazardous or damaging water levels e.g. because of variations in elevation across the city, and multi-story residential buildings or other infrastructure may provide refuge and prevent displacement. In contrast, a small village may get completely flooded quickly, offering little for its inhabitants to take refuge, and making it much more likely that most or all of its population may be displaced. These physical aspects concern fine-scale variations in exposure, which our data cannot distinguish, and which are thus subsumed in our vulnerability metric. In terms of social and economic reasons, rural areas tend to be poorer, with lower structural resilience of buildings, and to be neglected or treated subordinately by centralized government, resulting in higher vulnerability against floods and other disasters. For example, levees that protect larger settlements may lead to even higher flood levels for neighboring or downstream, smaller settlements. Rural areas may also lack economies of scale, as cities can afford much larger emergency response capacities such as professional fire brigades^{22,39}. Resilience against floods and other disasters differs markedly between rural and urban counties in the USA³⁹; such differences are likely to be more pronounced in less wealthy countries.”

In addition, Lines 212–214 state:

“However, such a bias would imply that we overestimate vulnerability in urban areas, and thus our finding of higher vulnerability in rural areas compared to urban areas remains robust.”

I recommend revising this sentence to explicitly indicate that the finding applies to fluvial and coastal floods.

We have revised the sentence:

“We recall that the extent of urban floods may be underestimated e.g. when short-lived or small features such as flash floods or flooded streets are missed by satellite imagery³³; however, such a bias would imply that we overestimate vulnerability in urban areas, and thus our finding of higher vulnerability to displacement from fluvial and coastal flooding in rural areas compared to urban areas remains robust.”

Terminology of income

GDP per capita is not a measure of household or individual income, and using the term “income” — especially at event-based or sub-national levels — may be misleading.

For example, in the Discussion section:

- Lines 184–186: “Our results show that the variance in flood-displacement vulnerability is better explained by factors other than average income levels alone (Fig. 2, top).”
- Lines 289–292: “The result that GDP per capita is not the most important predictor at sub-national level either suggests that, for similar reasons, income levels – at least as measured with available data products – may inappropriately capture vulnerability even when averaged over smaller areas.”

In both cases, “income levels” may be misinterpreted as referring to individual-level income rather than the aggregate GDP per capita used in the analysis. I recommend clarifying this by explicitly stating, for example, “aggregate income levels as measured by GDP per capita” or another related phrase, to avoid ambiguity.

As noted above, the title also uses “income levels”, which would also need to be rephrased.

We have clarified this in these and other places throughout the manuscript:

“Our results show that the variance in flood-displacement vulnerability is better explained by factors other than aggregate income levels (as measured by GDP per capita) alone (Fig. 2, top).”

“The result that GDP per capita is not the most important predictor at sub-national level either suggests that, for similar reasons, aggregate income levels – at least as measured with available data products – may inappropriately capture vulnerability even when averaged over smaller areas.”

“GDP per capita, though widely used to characterize socioeconomic status, may thus be a poor measure of vulnerability to flood-induced displacement. One reason may be that communities are very heterogeneous, and the poorest and most vulnerable parts of the population may not show up in average income levels as measured by GDP per capita, whereas factors like infant mortality are sensitive to the presence of marginalized, impoverished, or undersupplied subpopulations.”

“Thus, on the one hand, our study confirms a negative and non-linear association between aggregate income levels and vulnerability also with respect to displacement.”

“This means that high-income places tend to be less vulnerable than low-income places, but there is a lot of variation in vulnerability in both the low-income and the high-income range unexplained by income levels as measured by GDP per capita.”

Reviewer #2 (Remarks to the Author):

I have read the revised flood study and the revision memo with considerable interest. I believe the authors have satisfactorily addressed most of the issues raised by the reviewers in the previous round. The study has a clear if perhaps narrow contribution to the literature and in my mind warrants publication, pending responding to a few minor issues.

I still find the presentation of the study's novelty in the introduction overstated. Unlike what a reader would infer from reading lines 85 onwards, this study is not the first to use (a) remotely sensed flood data, (b) subnational displacement data, or (c) multivariate models. This concern was raised also in the initial round so I am surprised that the revision hasn't added clarity here. I encourage the authors to rather play up their contribution of using the best available data on observed floods at global scale (GFD) *in combination with* [presumably] the best available subnational displacement estimates (IDMC). This approach is unique and valuable, even if it comes at a cost of having a smaller empirical sample than previous global studies.

We have revised the paragraph as follows:

“Our combination of the best available global flood observation data with the most complete and detailed global displacement estimates is unique compared to previous global studies of flood vulnerability^{21,27,28}. Our main methodological choices are motivated as follows. First, we use remote-sensing, rather than modeled, flood hazard data to warrant consistency and avoid model uncertainty²⁹, providing more accurate exposure estimates for each flood. Second, we use geocoded displacement information on level-1 or level-2 subnational administrative units (e.g. provinces or districts) for a finer resolved analysis than at national level²¹. Thus, we can identify the local context of displacement events, and address variations in displacement vulnerability not only across, but also within, countries. Third, as opposed to many previous studies that have focused on single predictors of flood impacts, such as national income or population size, we choose a multivariate analysis. Drawing from a larger set of plausible predictors, and using random forest regression, our analysis can also account for non-linear effects of, and interactions between, these predictors. Finally, using the vulnerability ratio as the target variable controls for the expected close association between exposure and displacements prior to the regressions. This narrows the distribution of the target variable (Supplementary Fig. S1) and makes sure we estimate predictor effects on vulnerability rather than on exposure. The third and fourth aspects especially distinguish our work from a recent study that estimated displacements from a smaller number of local-level independent variables in a linear regression framework¹⁸.”

Table 1 and Fig. 2: It would be good to also indicate direction of effects for the predictors by adding signs, arrows, or different colors for positive vs. negative effects.

We now indicate the direction of effects using different colors in Fig. 2, and explain them in the caption:

Figure 2. Feature importance in the event-level analysis (top; $n=303$) and the country-level analysis (bottom; $n=72$), measured by the median decrease in R^2

after randomizing. Results relate to the test data using leave-one-out cross-validation. Each box plot represents all models using the predictor of interest; up to three predictors are used per (mixed effects) random forest model. Vertical line, box, whiskers, and circles indicate the median, interquartile range, 1.5 times the interquartile range, and outliers, respectively. Only models with $R^2 \geq 0.01$ are included. Box color indicates whether an increase in the predictor value is estimated to have an increasing (orange), decreasing (purple), or non-monotonic or ambiguous effect (grey) on vulnerability (cf. Fig. 3 and 4).

While revising the figure, we realized that we had failed to update the top panel of this figure to the latest version, as of the previous revision. We sincerely apologize for this. The current figure – as shown above and in the enclosed final revision of the manuscript – includes the updated version of the top panel; the main difference to the one included in the previous revision is that event duration is ranked seventh most important predictor, not fourth-most important. Otherwise, the predictor ranking has not changed, and the individual boxplots have changed only marginally, with no effect on any of the results or conclusions discussed in the paper. In fact, the Results section already discusses critical infrastructure as the fourth-most important predictor; we had simply failed to update the figure accordingly.

Line 236: My interpretation of the elevation effect is opposite: the marginal effect is highest in flat terrain (floodplains?), drops markedly as soon as some topography is introduced, and thereafter grows only slowly with degree of ruggedness. Consider revising.

We had already pointed out this observation at the end of the paragraph: “The partial dependence plot also indicates increased vulnerability at very low elevations, which usually have flat terrain and therefore might be associated with longer flood duration; this observation is based only on few samples and may be less reliable (Fig. 3).” We have further revised this to provide some additional considerations:

“The partial dependence plot also indicates increased vulnerability at very low elevations, although this observation is based only on few samples and may be less reliable (Fig. 3). These areas below approx. 10m above sea level are mainly coastal areas which globally are often densely populated and susceptible to coastal flooding; with their flat terrain, they may be associated with longer flood duration on average than more rugged areas.”

Discussion section (lines 165-350): the write-up is rather technical and repetitive (dare I say boring?) and disconnected from the real world. Some references to illustrative cases along the way would help substantiating the analysis. More could be done also in terms of exemplifying the estimated effects: how much would Y_{hat} change for a given change in X , all else constant?

Recognizing the space limits, we have tried to improve our Discussion section through the following additions:

“The remaining effect of population density on vulnerability, highlighted by our models, indicates residents of rural areas on average suffer higher displacement risk than their urban counterparts, for a given hazard. Our models estimate two orders of magnitude difference in vulnerability between sparsely and densely populated areas, all else equal; which would imply a large potential for risk reduction from better protecting rural populations.”

And:

“Again, we find that GDP per capita is relatively unimportant in the country-level models. GDP per capita, though widely used to characterize socioeconomic status, may thus be a poor measure of vulnerability to flood-induced displacement. One reason may be that communities are very heterogeneous, and the poorest and most vulnerable parts of the population may not show up in average income levels as measured by GDP per capita, whereas factors like infant mortality are sensitive to the presence of marginalized, impoverished, or undersupplied subpopulations. The marginal effect of the infant mortality indicator on vulnerability is highly nonlinear, highlighting the usefulness of models like random forest that do not impose a-priori linear relationships. For the real world, our results suggest reducing poverty and improving living conditions for the most vulnerable parts of the population might, among countless other benefits, effectively reduce displacement risk.”

More could be said about the modest influence of the FLOPROS predictor, which intuitively should have a powerful effect – if current flood protection measures are effective in reducing displacement. The discussion section (e.g., lines 391-404) mentions the importance of physical protection measures, but fails to connect this discussion with the FLOPROS estimate.

We have added explanation of the FLOPROS predictor in the Results section:

“The remaining predictors show mostly small or indeterminate marginal effects (Fig. 3 and Supplementary Fig. S9), which is consistent with their low feature importance ranking. This includes a measure of flood protection standards (FLOPROS), which in our context does not represent the effectiveness of flood prevention (we only study floods which were not prevented) but the possibility that higher flood protection standards may also be associated with stronger flood emergency response capacities. However, according to our analysis, this measure is of low importance in explaining displacement vulnerability; which may also be related to the high uncertainty of protection standard estimates in many parts of the world⁴⁵. ”

And in the Discussion section:

“High infant mortality rates are also related to the marginalization of affected communities⁵⁷, which is associated with higher flood risk⁵⁸, caused by, for example, a lack of early warning systems, physical protection measures (which are not well measured by FLOPROS in many Global South countries), or official emergency and recovery support.”